# The role of experience in prioritizing hippocampal replay

Marta Huelin Gorriz [1], Masahiro Takigawa [1] & Daniel Bendor [1] ✉

During sleep, recent memories are replayed by the hippocampus, leading to their consolidation, with a higher priority given to salient experiences. To examine the role of replay in the selective strengthening of memories, we recorded large ensembles of hippocampal place cells while male rats ran repeated spatial trajectories on two linear tracks, differing in either their familiarity or number of laps run. We observed that during sleep, the rate of replay events for a given track increased proportionally with the number of spatial trajectories run by the animal. In contrast, the rate of sleep replay events decreased if the animal was more familiar with the track. Furthermore, we find that the cumulative number of awake replay events occurring during behavior, influenced by both the novelty and duration of an experience, predicts which memories are prioritized for sleep replay, providing a more parsimonious neural correlate for the selective strengthening of memories.

Memory storage goes through a two-stage process in the brain[1]. First behavioral episodes are encoded by the hippocampus, and then during sleep, these memories are consolidated, with information initially stored in the hippocampus gradually transformed into a stable, long-term memory[2]. Neurons in the rodent hippocampus, commonly referred to as place cells, are spatially tuned and fire action potentials when the animal is positioned within a specific region of the environment, known as a place field[3]. As the rodent runs along a spatial trajectory through multiple place fields, the corresponding place cells fire in a temporal sequence. During offline states such as quiet restfulness and non-REM sleep, these same place cells spontaneously reactivate this sequential pattern, reinstating a neural memory trace of the spatial trajectory previously traversed[4,5]. The link between hippocampal replay, memory consolidation, and sleep is supported by extensive experimental evidence demonstrating (1) a cortico-hippocampal dialog during sleep, leading to coordinated replay activity, and (2) a memory enhancement (or disruption) following a positive (or negative) manipulation in sleep and/or replay[6–18].

However, not all memories are equally valuable for an animal's survival. For example, the location of a new food cache should take precedence over other regions in the environment that provide no benefit to the animal and are less likely to be visited in the future. As consolidation is a time-limited process, there is an information bottleneck for memory storage and a need for efficiency. It is hypothesized that the brain prioritizes the consolidation of important and novel memories while delaying or even triaging memories that are less important to consolidate, leading to their eventual forgetting[6,19]. Furthermore, as memories get more strongly consolidated, the need to continue consolidating these memories should decrease, providing more opportunities for the consolidation of newer memories not yet benefited from this process. Data in humans indicate that memory prioritization is a consequence of an experience's salience, which can take multiple forms, including emotional[20], repetition[21], reward[22], or even perceived future relevance, such as simply informing participants after training that the task will be later tested[23]. This ability of the brain to bias memory processing towards more salient experiences requires sleep; if subjects remained awake following training on a behavioral task for a similar time period, no bias in-memory processing is observed[23]. Furthermore, if sleep does not occur within the 24-h period following the behavioral task, any saliency bias in memory is permanently lost, even with two nights of recovery sleep following sleep deprivation[24]. These data suggest that memories can be effectively prioritized or triaged, but only within a limited time window. How this is accomplished by the brain, specifically how the hippocampus tags salient memories during learning for later prioritization during this sleep window for consolidation is unknown.

[1]Institute of Behavioural Neuroscience (IBN), University College London (UCL), London WC1H 0AP, UK. ✉e-mail: d.bendor@ucl.ac.uk

## Results

We examined hippocampal replay during post-behavior sleep periods after male rats (*n* = 4) ran spatial trajectories on two linear tracks (Fig. 1A). Following a 1-h sleep period (PRE) in a quiet remote location (*rest box*) from the two tracks, the rat ran spatial trajectories on two novel tracks (RUN1), motivated by a small liquid reward delivered at each end of the track, every lap. During the first exposure to the novel tracks, the two tracks differed in the number of laps the rat was permitted to run (16 laps for track 1, and between 1–8 laps for track 2). Next, the rat was returned to the rest box for a 2 h rest/sleep session (POST1) and then returned back to the same two linear tracks to run additional spatial trajectories (RUN2). During the re-exposure to the two linear tracks, the amount of time the rat spent running on each track was fixed (approximately 15 min/track). However, during RUN2, the familiarity of the two tracks presumably differed as a consequence of the number of laps previously run in RUN1 (i.e., always more laps were previously run on track 1 than track 2). Finally, following RUN2,

the rat had an additional rest/sleep session in the rest box for 1 h (POST2). This experiment was repeated across 5 days, each with a different lap condition on track 2 (1, 2, 3, 4, or 8 laps) during RUN1. To generate novel environments each day, the track geometries and textures were changed, and the two tracks were repositioned in the room with new cues (Fig. 1B). The configurations of the room (e.g., track geometry and placement, barriers for visual occlusion) and lap conditions were pseudo-randomly ordered for each subject. During RUN1, the additional laps on track 1 relative to track 2 (Fig. 1C and Fig S1A), led to an increase in the total time spent immobile (speed <4 cm/s) (Fig. 1D and Fig. S1B, two-sided Wilcoxon signed-rank test, *p* = 0.00013, *n* = 19) and running (Fig. 1E and Fig. S1C, two-sided Wilcoxon signed-rank test, *p* = 0.00013, *n* = 19). However, no difference was observed between the two tracks during the re-exposure session (RUN2, two-sided Wilcoxon signed-rank test, *p* = 0.78 for total time spent immobile and *p* = 0.26 for time spent mobile, *n* = 19). Furthermore, across all protocols, the average running speed of the animal

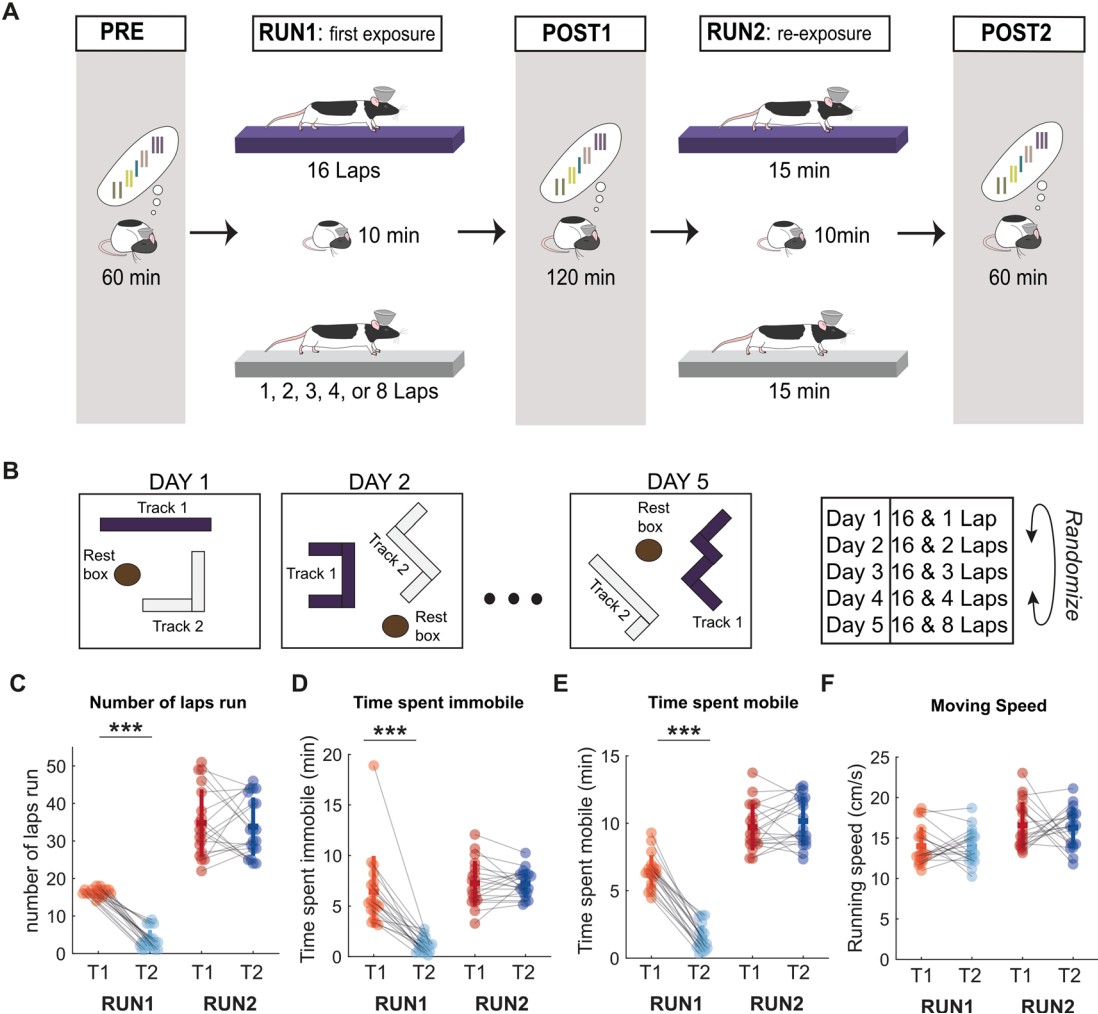

**Fig. 1 | Behavioral manipulation of spatial experience. A** Experimental design. A given session started with a 1-h rest period in a remote rest box (PRE) followed by the first exposure to two novel linear tracks (RUN1), differing in the number of laps allowed to explore (16 laps on Track 1 and between 1–8 laps on Track 2). After RUN1, the rat was allowed to rest for 2 h in the rest pot (POST1) before being re-exposed to both tracks (RUN2), where rat was allowed run on each track for approximately 15 min. The session ended with a 1-h rest session (POST2). **B** The configuration of both tracks and the rest box (including track geometry and placement) as well as the number of laps ran on Track 2 during RUN1 were pseudo-randomized for each rat. **C**–**F** Summary of animal behavior during RUN1 and RUN2. Data points from track 1 (T1) and track 2 (T2) during the first exposure (RUN1) are

indicated using orange and light blue, respectively. Data points from track 1 and track 2 during the second exposure (RUN2) are indicated using red and dark blue. Error bars are presented as mean ± SD for panels (**C**–**F**). ****p* < 0.001, two-tailed Wilcoxon signed-rank test. **C** Number of laps run for each protocol. ****p* = 0.00013 for T1 vs. T2 during RUN1 and *p* = 0.72 for T1 vs. T2 during RUN2. **D** Time spent immobile for each protocol. ****p* = 0.00013 for T1 vs. T2 during RUN1 and *p* = 0.78 for T1 vs. T2 during RUN2. **E** Time spent mobile for each protocol. ****p* = 0.00013 for T1 vs. T2 during RUN1 and *p* = 0.26 for T1 vs. T2 during RUN2. **F** Moving speed for each protocol. *p* = 0.94 for T1 vs. T2 during RUN1 and *p* = 0.97 for T1 vs. T2 during RUN2. *n* = 19 sessions from 4 rats.

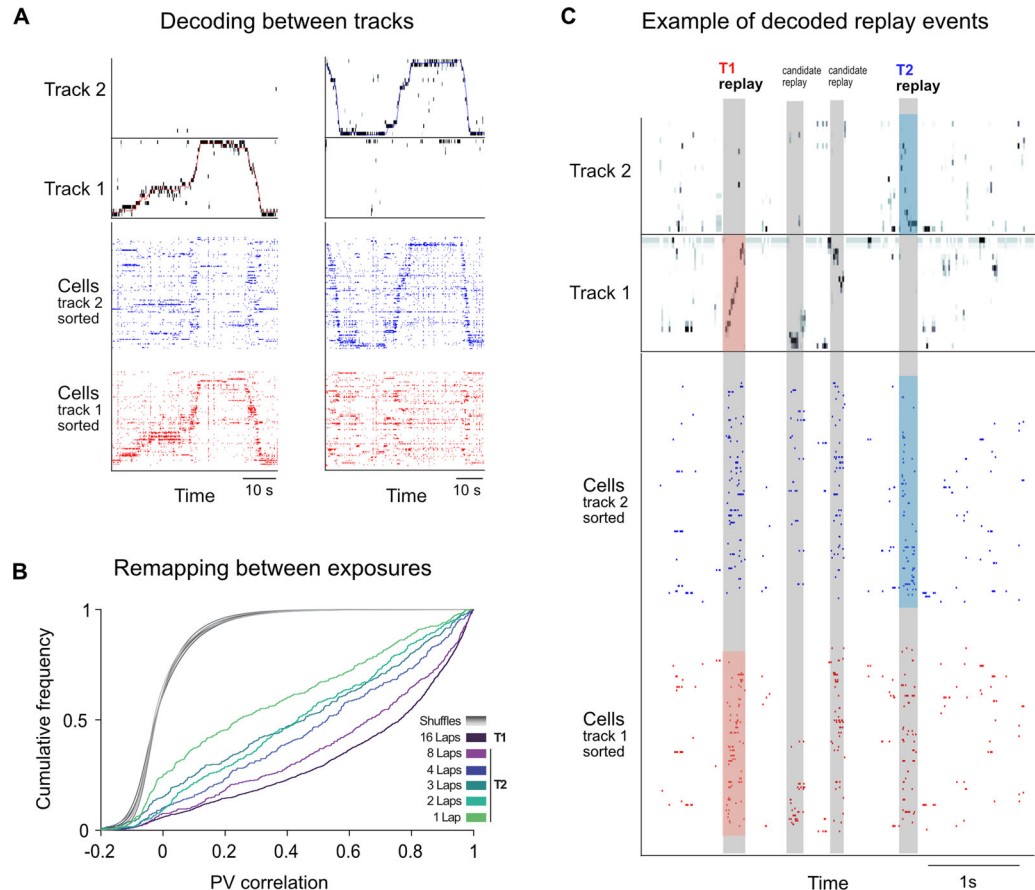

**Fig. 2 | Hippocampal remapping across tracks and representational stability across exposures. A** Raster plot of cells spiking activity sorted by their peak firing rate on the track, and the associated decoded posterior probability for example laps from both tracks. **B** Cumulative frequency of the population vector (PV) correlations between place field maps during RUN1 and RUN2. Lines with different colors were used to indicate data from different experimental protocols: dark purple for T1 with 16 laps, magenta for T2 with 8 laps, blue-violet for T2 with 4 laps, blue-green for T2 with 3 laps, jade green for T2 with 2 laps and green for T2 with 1 lap. Gray lines represent the distributions of the shuffled data. **C** Example neuronal activity and decoded posterior probability for replay events during POST1 from Rat 2 with a 16-4 laps protocol on track 2 during RUN1.

between tracks remained similar (Fig. 1F and Fig. S1D, two-sided Wilcoxon signed-rank test, $p = 0.94$ for RUN1 and $p = 0.97$ for RUN2, $n = 19$).

Using chronically implanted microdrives[25–27], we recorded from large ensembles of place cells during our behavioral task and the corresponding sleep sessions. We observed that place cells globally remapped between tracks[27,28], while having similar place fields on the same track between RUN1 (first exposure) and RUN2 (re-exposure) (Fig. 2A, B and Fig. S2). The majority of sessions had at least 50 place cells per track (Fig. S3A). Using a naïve Bayes decoder, we reconstructed the rat's position on each track from the firing rates of place cells during RUN1 and RUN2, and using this same approach we reconstructed the virtual spatial trajectories within replay events across the entire recording session. Candidate replay events were selected based on a minimum duration >= 100 ms, and a $z$-score threshold of 3 for the smoothed multi-unit activity across all channels[27] (see Methods). Candidate replay events with a $z$-score > 3 for ripple-band power, and a statistically significant weighted correlation score ($p < 0.05$) for three different shuffle distributions[27,29] (see Methods) were considered significant replay events and included in our analysis (Fig. 2C). Replay events during POST1 and POST2 were classified as replay during putative sleep state (referred as *sleep* replay hereafter) if the replay occurred when the animals' mean moving speed within a 1-min time bin was lower than 4 cm/s and the $z$-scored MUA activity of the most active units (top one-third of the units in terms of total spike count) was above 0. Otherwise, they were classified as *rest* replay (see Methods).

We first examined how the number of laps run during RUN1 influenced the rate of replay (i.e., the number of replay events per unit time) that occurred during POST1, focusing on the first 30 min of cumulative sleep (see Methods). We selected the first 30 min of cumulative putative sleep as this timeframe represents the longest minimum duration that all animals attained in most sessions. In addition, for all animals, the first 30 min of cumulative sleep was distributed similarly across time, independent of the session condition. Using place fields in RUN1 to decode POST1 replay events, we observed that the rate of sleep replay increased with the number of laps run by the animal (Fig. 3A, B). On the first track, the rat always ran 16 laps, and correspondingly, sleep replay rates were similar across protocols and generally higher than track 2 (for which the rat always ran fewer laps). On track 2, replay rates decreased as the number of laps was reduced across protocols. We observed a significantly higher sleep replay rate in POST1 for track 1 compared to track 2 across all protocols (Fig. 3B, T1 and T2 mean ± SD: T1 = 0.0310 ± 0.01 events/s, T2 = 0.0185 ± 0.0077 events/s, two-sided Wilcoxon signed-rank test, $p = 0.0002$, $n = 19$). This difference was also observed in the proportion of active place cells in replay events of track 1 and track 2 during POST1 (Fig. S3B).

While both tracks were novel to the rat in RUN1, this was no longer the case in RUN2 when rats were re-exposed to the same tracks. While the time spent on each track was fixed (approximately 15 min each), the tracks now varied in their familiarity, given that the rat always has had more previous experience (number of laps) on track 1 relative to

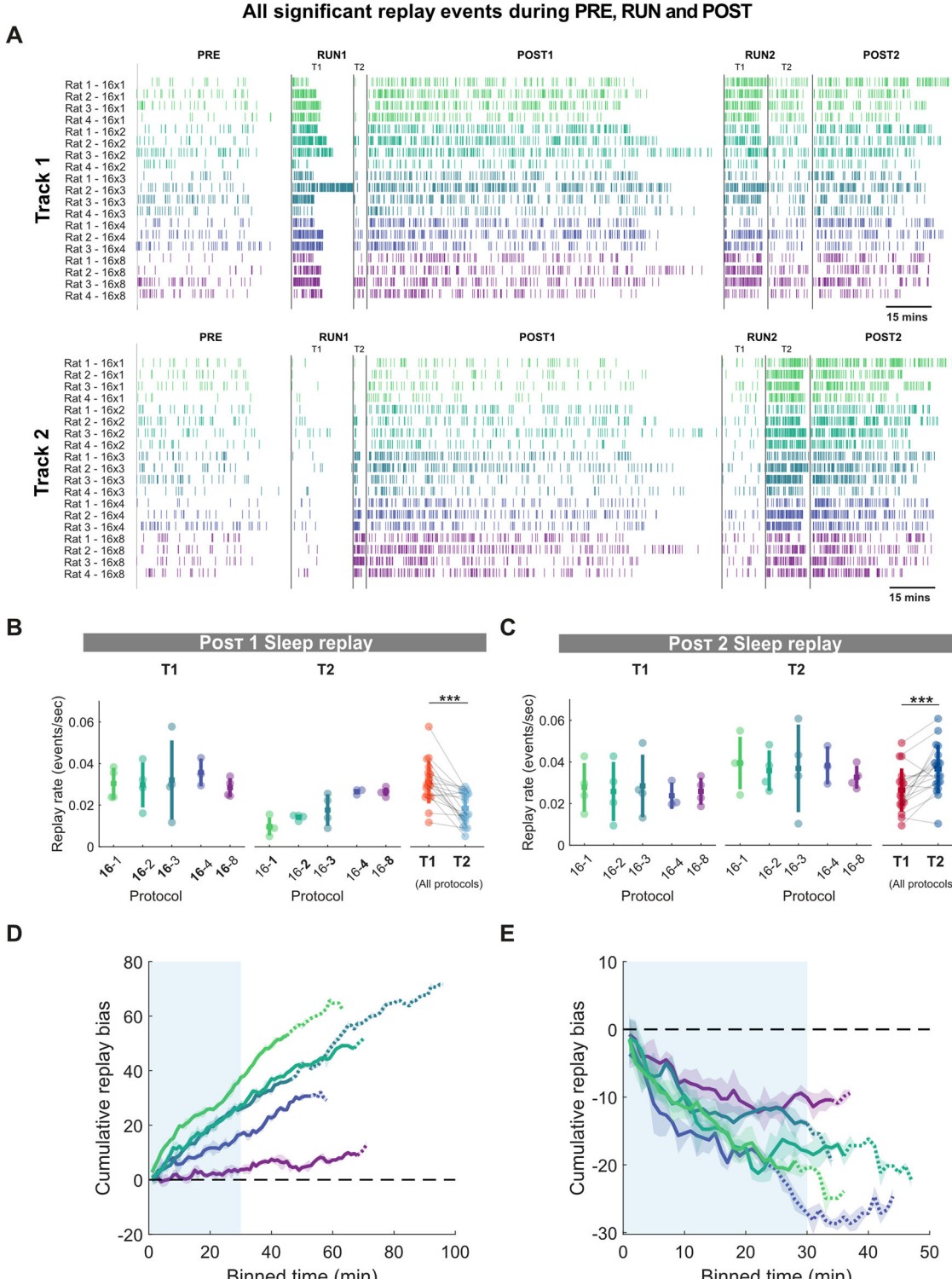

**Fig. 3 | Hippocampal sleep replay increases with repetition but decreases with familiarity. A** Raster plot of significant replay events for track 1 (top) and track 2 (bottom) during the pre-exposure rest period (PRE), the first exposure to track 1 (RUN1 T1) and track 2 (RUN1 T2), the rest period following the first exposure (POST1), the re-exposure to track 1 (RUN2 T1) and track 2 (RUN2 T2) and the final rest period (POST2). **B, C** Rate of sleep replay for track 1 and track 2 during first 30 min of cumulative sleep of POST1 (**B**) and POST2 (**C**). For POST1, the lap number associated with each track during RUN1 was highlighted in bold. For POST2, the lap number was not highlighted as the time spent on the two tracks was nearly identical. Each datapoint represents the mean sleep replay rate within a session, color-coded according to the experimental protocol and track identity (same as Fig. 1C–F and Fig. 2B). ***p = 0.0002 for T1 vs. T2 POST1 sleep replay (**B**), ***p = 0.002 for T1 vs. T2 POST sleep replay (**C**), two-tailed Wilcoxon signed-rank test. n = 19 sessions from 4 rats. Error bars are presented as mean ± SD for panels (**B, C**). **D, E** Cumulative sleep replay bias across protocols during POST1 (**D**) and POST2 (**E**). The solid line and the shaded region represent the mean ± standard deviation across all sessions for a given protocol. Color scheme according to the experimental protocol (same as Fig. 2B). The solid line becomes a dashed line when less than half of the animals are contributing to the data at each time point. The light blue box outlines the first 30 min of cumulative sleep time windows used for analysis in **B, C**.

track 2. Using place fields from RUN2 to decode POST2 replay events, we observed that replay rates in the first 30 min of cumulative sleep were significantly higher for track 2 compared to track 1, suggesting that the more familiar track replayed less during periods of putative sleep (Fig. 3C, T1 and T2 mean ± SD: T1 = 0.0265 ± 0.010 events/s, T2 = 0.0366 ± 0.011 events/s, two-tailed Wilcoxon signed-rank test $p = 0.002$, $n = 19$). This difference was also observed in the proportion of active place cells in replay events of track 1 and track 2 during POST2 (Fig. S3B).

To help control for potential changes in replay rates across sessions and subjects, we next calculated the track replay bias, by subtracting the number of track 1 replay events from the number of track 2 replay events within a specified time window. Calculating the cumulative track replay bias over time, we observed that the difference in the number of replay events between track 1 and track 2 grew linearly in the positive direction (track 1 > track 2) during POST1 sleep periods. This suggests that the replay bias towards track 1 was present at the start of the sleep session and was maintained throughout POST1, beyond the 30 min of cumulative sleep that was used in our main analysis (Fig. 3B, D). Several labs have reported a decay of replay within the first hour of sleep[8,30–32], however, more recently replay has been reported to last for hours, and even days, after the experience[33–36]. We observed that the rate of sleep replay for both track 1 and track 2 slowly decayed (with a similar slope) during POST1 (Fig. S4A, B). This indicates that a constant bias in the rate of sleep replay between tracks during POST1 can nevertheless still be maintained in spite of an overall decrease in sleep replay rates over time. Interestingly, the decay in sleep replay rates differed in slope between tracks during POST2 (Fig. S4C, D), suggesting that decay rates are not always fixed and may depend on behavioral parameters in the task (e.g., familiarity).

We next examined how the track replay bias was affected by familiarity during POST2, where familiarity increased with the amount of prior exposure (number of laps) run on the track during RUN1. We observed a negative track replay bias for periods of sleep across POST2, indicating that track 2 replayed more than track 1 throughout POST2 (albeit over a shorter sleep session), with the cumulative change in bias between track 2 and track 1 more similar between protocols (Fig. 3C, E). Together, these results suggest that both repetition and familiarity influence sleep replay resulting in (**1**) higher replay rates with more experience on a novel track (more laps), and (**2**) lower replay rates with greater familiarity (i.e., more prior experience on the same track). Interestingly, we observed a similar qualitative relationship for repetition and familiarity with *rest* replay events (Fig. S5), however, track differences were only statistically significant in POST2 (POST1 T1 and T2 mean ± SD: T1 = 0.0186 ± 0.0074, T2 = 0.0160 ± 0.0081, two-tailed Wilcoxon signed-rank test $p = 0.3164$, $n = 19$; POST2 T1 = 0.0100 ± 0.0056, T2 = 0.0188 ± 0.0072, two-tailed Wilcoxon signed-rank test $p = 0.0010$, $n = 19$).

We next investigated whether any additional factors that varied between tracks could influence the rate of replay. Although we were able to decode the trajectories accurately across all protocols (Fig. S2), the median decoding error generally decreased with the number of laps run by the animal on a novel track. As an alternative approach, we measured sleep replay during POST1 using place fields obtained in RUN2, which were fully stabilized after 15 min of running on each track. Using this approach, we observed the identical trend of a higher rate of track 1 sleep replay (relative to track 2) during POST1 (Fig. S6A). Given the possibility that the place cell representations may partially remap between RUN1 and RUN2, we also quantified sleep replay during POST1 using place fields calculated from the final (single) lap for both track 1 and track 2 regardless of the total laps ran. The result remained consistent with the main finding (Fig. S6B). Because we were limited in the number of sessions that could be recorded from each animal, we used a fixed protocol with the rat always running more laps on track 1 compared to track 2, and with track 1 always being the first track used

in the re-exposure. While it is possible that sleep replay rates increase with the recency of the behavioral episode to the sleep session (i.e., higher replay rates for track 2), previous replay data comparing replay between two novel tracks with a similar duration of experience suggests that this is not a significant effect[27].

How does the hippocampus choose which memories to prioritize for replay, given that the cues important for this decision (e.g., reward, repetition, and familiarity) occur during the behavioral episode, and are no longer present later during sleep? We next explored candidate neural correlates during behavior that could predict the difference in sleep replay rates observed during both POST1 and POST2. Both learning and memory consolidation are postulated to rely on the phenomenon of theta sequences, the ordered firing of place cells occurring every theta cycle, encoding spatial trajectories sweeping from behind to in front of the animal's current position[37]. More recently, it has been shown that impaired theta sequences result in degraded sleep replay, suggesting that theta sequences may be necessary for the initial formation of memory traces[17]. Alternatively, the repeated ordered firing of place cells during behavior, postulated to be necessary for sleep replay to later occur, could instead be produced during awake replay. This form of replay is qualitatively similar to sleep replay but typically occurs during behavioral episodes and outside periods of locomotion-driven theta activity when the animal is resting, grooming, or consuming reward[38,39]. During awake replay, a spatial trajectory within the local environment typically reactivates, however, although less frequent, remote replay of an experience outside of the current environment is also possible[40–43]. Awake replay may play multiple functional roles, with evidence supporting both goal-directed behavior[44,45] and memory function[46,47].

First, to examine the role of theta sequences in modulating the rate of sleep replay, we computed the Bayesian decoded theta cycle normalized by the animal's position and duration of the theta cycle, and observed strengthening of the sequential structure (the decoded position gradually shifting from behind to in front of the animal) as the number of laps increased, while theta sequences for both tracks during RUN2 were qualitatively similar (Fig. 4A). The number of theta sequences (weighted correlation, $p < 0.05$, see Methods) was significantly higher for track 1 compared to track 2 during RUN1 (Fig. 4B, T1 and T2 mean ± SD: T1 = 1546 ± 647 theta sequences, T2 = 263 ± 205 theta sequences, two-tailed Wilcoxon signed rank $p = 0.0001$, $n = 19$), matching the pattern of track 1 and track 2 sleep replay during POST 1. However, we did not observe a significant difference in the number of theta sequences between track 1 and track 2 for RUN2 (Fig. 4C, T1 and T2 mean ± SD: T1 = 1751 ± 753 theta sequences, T2 = 1702 ± 850 theta sequences, two-tailed Wilcoxon signed-rank test $p = 0.5732$, $n = 19$).

We next examined if local awake replay during the behavioral task influenced the rate of sleep replay (see Methods). We observed that the rate of awake replay was similar across protocols in RUN1, without a statistically significant difference between track 1 and track 2, albeit with the rate of awake replay during a single lap on track 2 being qualitatively much lower than other protocols (Fig. 5A, T1 and T2 mean ± SD: T1 = 0.13 ± 0.042 events/s, T2 = 0.12 ± 0.070 events/s, two-tailed Wilcoxon signed-rank $p = 0.60$, $n = 19$). However, the rate of awake replay was higher for track 2 compared to track 1 during RUN2 (Fig. 5B, T1 and T2 mean ± SD: T1 = 0.086 ± 0.033, T2 = 0.13 ± 0.042, two-tailed Wilcoxon signed-rank $p = 0.0038$, $n = 19$), similar to the relationship of track 1 and track 2 sleep replay during POST2.

Unlike theta sequences which were strengthened with familiarity, local replay rates on the track decreased with increasing familiarity. We quantified this by calculating the population vector correlation between place cells from the same track between RUN1 and RUN2 and found a significant regression with the rate of local awake replay during RUN2 (Fig. 5C, $R2 = 0.273$, $p = 4.49 \times 10^{-4}$). In other words, the more the place cells remapped (or were not stable) between RUN1 and RUN2 for a given track, which was more prevalent with less prior experience

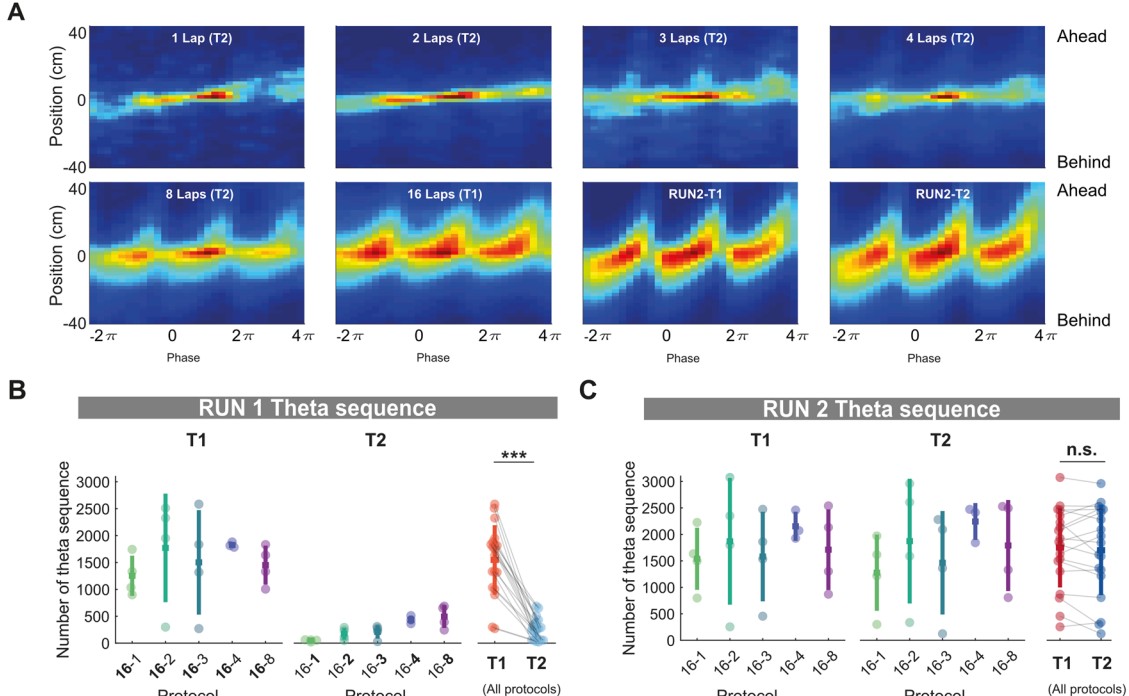

**Fig. 4 | Number of theta sequences increases with the repetition of trajectories but are insensitive to contextual novelty and familiarity. A** Averaged theta sequences per protocol. Each plot represents the averaged decoded probabilities of three consecutive theta sweeps over 80 cm. The sequences are centered in the trough of the theta cycle and the animal's current position on the *x*-axis and the *y*-axis, respectively. The hotness of the color indicates the magnitude of the decoded probabilities. **B**, **C** Number of theta sequence on track 1 and track 2 during RUN1 (**B**) and RUN2 (**C**) for each protocol. The lap number associated with each track during during RUN1 was made in bold. The lap number was not highlighted as the time spent on the two tracks was nearly identical. Each datapoint represents the mean theta sequence for a given track within a session, color-coded according to the experimental protocol and track identity (same as Fig. 1C–F and Fig. 2B). ***$p = 0.0001$ for T1 vs. T2 RUN1 theta sequence (**B**), $p = 0.57$ for T1 vs. T2 RUN2 theta sequence (**C**), two-tailed Wilcoxon signed-rank test. $n = 19$ sessions from 4 rats. Error bars are presented as mean ± SD for panels **B**, **C**.

during RUN1, the higher the rate of local awake replay. These results support previous work reporting that more reactivations occur during behavioral episodes for novel environments compared to familiar ones[48–50].

Thus far, theta sequence number and awake replay rate were unable to explain the combined effect of repetition and familiarity/novelty on sleep replay. However, we speculated that the cumulative number of awake replay events should differ between tracks in RUN1, much like theta sequences, as a consequence of time spent on the track. Furthermore, the cumulative number of awake replay events should also be sensitive to familiarity during RUN2, as a consequence of awake replay rates decreasing with familiarity, and a similar time spent on each track (15 min each). We observed that for RUN1, awake replay number was significantly higher for track 1 compared to track 2 (Fig. 6A, T1 and T2 mean ± SD: T1 = 53.1 ± 35.5, T2 = 9.5 ± 8.5, two-tailed Wilcoxon signed-rank test $p = 0.0001$, $n = 19$), while the opposite was true for RUN2, with track 2 having significantly more awake replay events than track 1 (Fig. 6B, T1 and T2 mean ± SD: T1 = 37.5 ± 17.0, T2 = 56.1 ± 20.8, Signed rank test $p = 0.0013$, $n = 19$). Thus, unlike our observation for theta sequences and awake replay rate, the cumulative number of local awake replay events RUN1 and RUN2 better mirrored the pattern of track 1 and track 2 sleep replay during both POST1 (T1 > T2) and POST2 (T1 < T2), respectively.

To further explore the role of theta sequence and awake replay in shaping sleep replay rate, we applied a linear regression analysis to identify the factors that could more universally predict the rate of sleep replay during both POST1 and POST2. Using a simple regression analysis, we found that the time spent on a track due to task manipulation (Fig. 7A, R2 = 0.347, $p = 1.3 \times 10^8$, $n = 76$), as well as all the three candidate neural correlates were predictive of the rate of sleep replay

(Fig. 7B–D, number of theta sequence: R2 = 0.390, $p = 1.02 \times 10^{-9}$, $n = 76$, rate of awake replay: R2 = 0.219, $p = 1.2 \times 10^{-5}$, $n = 76$, number of awake replay: R2 = 0.554, $p = 7.9 \times 10^{-15}$, $n = 76$). Extending our regression analysis to look more broadly at theta cycles and awake sharp-wave ripple (SWR) events (see Methods), the predictive relationships remained significant for both number of theta cycles and awake SWR events (but not awake SWR rate) (Fig. 7E–G, number of theta cycle: R2 = 0.338, $p = 2.22 \times 10^{-8}$, $n = 76$, rate of awake SWR events: R2 = −0.013, $p = 0.806$, $n = 76$, number of awake SWR events: R2 = 0.497, $p = 7.27 \times 10^{-13}$, $n = 76$). While all three metrics showed a statistically significant regression with sleep replay rate, it is plausible that these three neural correlates exert different degrees of influence on sleep replay, or alternatively may be a byproduct of any collinearity across behavioral protocols. Therefore, we next sought to examine the relative contribution of theta sequences and awake replay toward sleep replay rate using a mixed-effect linear regression analysis. The model used the three candidate neural correlates as well as the time spent on track as the fixed effects and included the subject id as the random effect to account for animal variability. All metrics were *z*-scored such that the standardized beta coefficient could be used to compare the relative weight of each fixed-effect term. We calculated the 95% confidence interval for the standardized beta coefficient such that the relative predictive power of the metric would be considered statistically insignificant when it overlapped with zero. We found that only the standardized beta coefficient of rate and cumulative number of awake replay were significantly greater than zero (Fig. 7H), highlighting the relative importance of awake replay in explaining sleep replay. If alternatively, we only used theta cycles and SWR events during behavior, metrics that rely on oscillatory activity but not place cell sequences (thus not sensitive to the size of the place cell

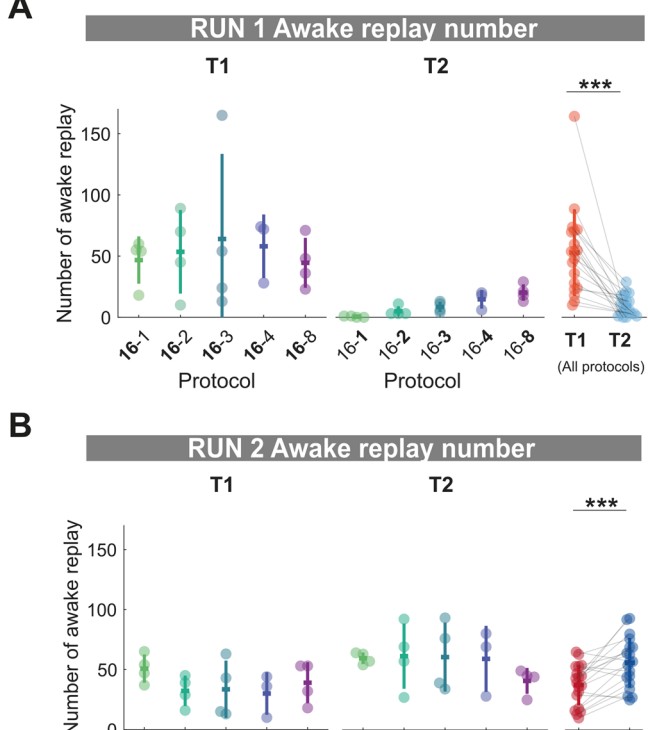

**Fig. 6 | Local awake replay number increases with repetition but decreases with familiarity. A**, **B** Number of local awake replay on track 1 and track 2 during RUN1 (**A**) and RUN2 (**B**) for each protocol. Each datapoint represents the mean number of local awake replay for a given track within a session, color-coded according to the experimental protocol and track identity (same as Fig. 1C–F and Fig. 2B).\*\*\**p* = 0.00013 for T1 vs. T2 RUN1 awake replay number, \*\*\**p* = 0.0013 for T1 vs. T2 RUN2 awake replay number, two-tailed Wilcoxon signed-rank test. *n* = 19 sessions from 4 rats. Error bars are presented as mean ± SD for panels **A**, **B**.

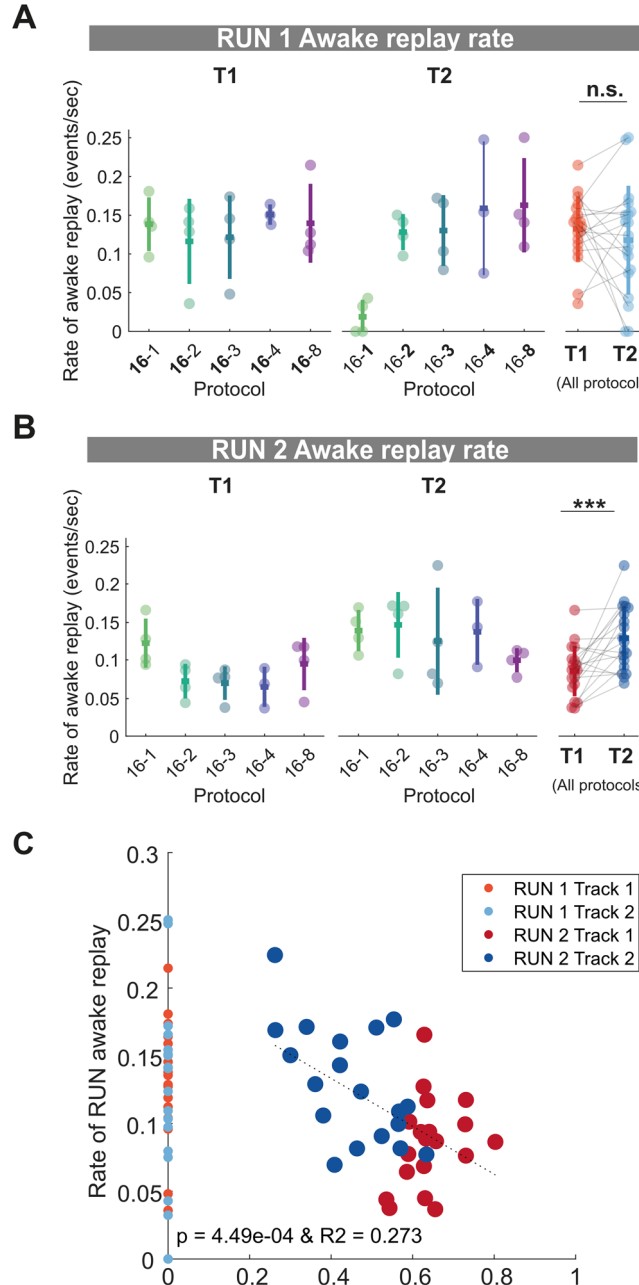

**Fig. 5 | Rate of local awake replay is sensitive to contextual novelty and familiarity. A**, **B** Rate of local awake replay on track 1 and track 2 during RUN1 (**A**) and RUN2 (**B**) for each protocol. Each datapoint represents the mean local awake replay rate for a given track within a session, color-coded according to the experimental protocol and track identity (same as Fig. 1C–F and Fig. 2B). *p* = 0.60 for T1 vs. T2 RUN1 awake replay rate (**A**), \*\*\**p* = 0.0038 for T1 vs. T2 RUN2 awake replay rate (**B**), two-tailed Wilcoxon signed-rank test). The lap number associated with each track during RUN1 is indicated in bold. In RUN2, the lap number is not highlighted as the time spent on the two tracks is similar. *n* = 19 sessions from 4 rats. Error bars are presented as mean ± SD for panels **A**, **B**. **C** Regression between the population vector correlation between exposures and the rate of local awake replay during RUN2. The rate of local awake replay during RUN1 for each session is plotted on the *x*-axis at position 0 for visualization purposes only. *n* = 38 datapoint from both tracks during RUN2 of 19 sessions (4 rats).

ensemble), we found that only the standardized beta coefficient for the cumulative number of awake SWR events was significantly greater than zero (Fig. 7I). Interestingly, in contrast to *sleep* replay, the rate and cumulative number of awake replay events but not theta sequences

were predictive of *rest* replay during the POST sessions (Fig. S7A–G). Using a mixed-effect regression analysis, awake replay number remained the best predictor of rest replay with a significant standardized beta coefficient (Fig. S7H). Therefore, while these results do not eliminate a possible role for theta sequences in sleep replay, especially when the environment was novel, the prioritization of offline sleep replay was more parsimoniously explained across our entire dataset by the cumulative number of awake replay events.

To further validate our observation that a higher number of local awake replay events during a behavioral episode resulted in a higher priority for this spatial trajectory to subsequently replay during sleep, we extended our analysis to individual place cells. We examined only place cells with place fields on both tracks, and asked whether the difference in the number of local awake replay events a given cell participates in (track 1 – track 2) during RUN, predicts the observed difference in sleep replay rates for that cell (track 1 – track 2) during the subsequent POST session. We observed a significant regression in 18 out of 19 POST1 sleep sessions and 16 out of 19 POST2 sleep sessions (Fig. S8), indicating that the more times a cell participates in the replay of a given track, the more likely that cell is to participate during the sleep replay of that track. To confirm that the result is not trivially explained by the overall firing rate on the track and during sleep or decoding accuracy, we further applied a decoding-independent analysis. Instead of using Bayesian decoding, we classified SWR events as either track 1 or track 2 reactivation events when the proportion of one of the track's selective place cells was 20% greater than the proportion

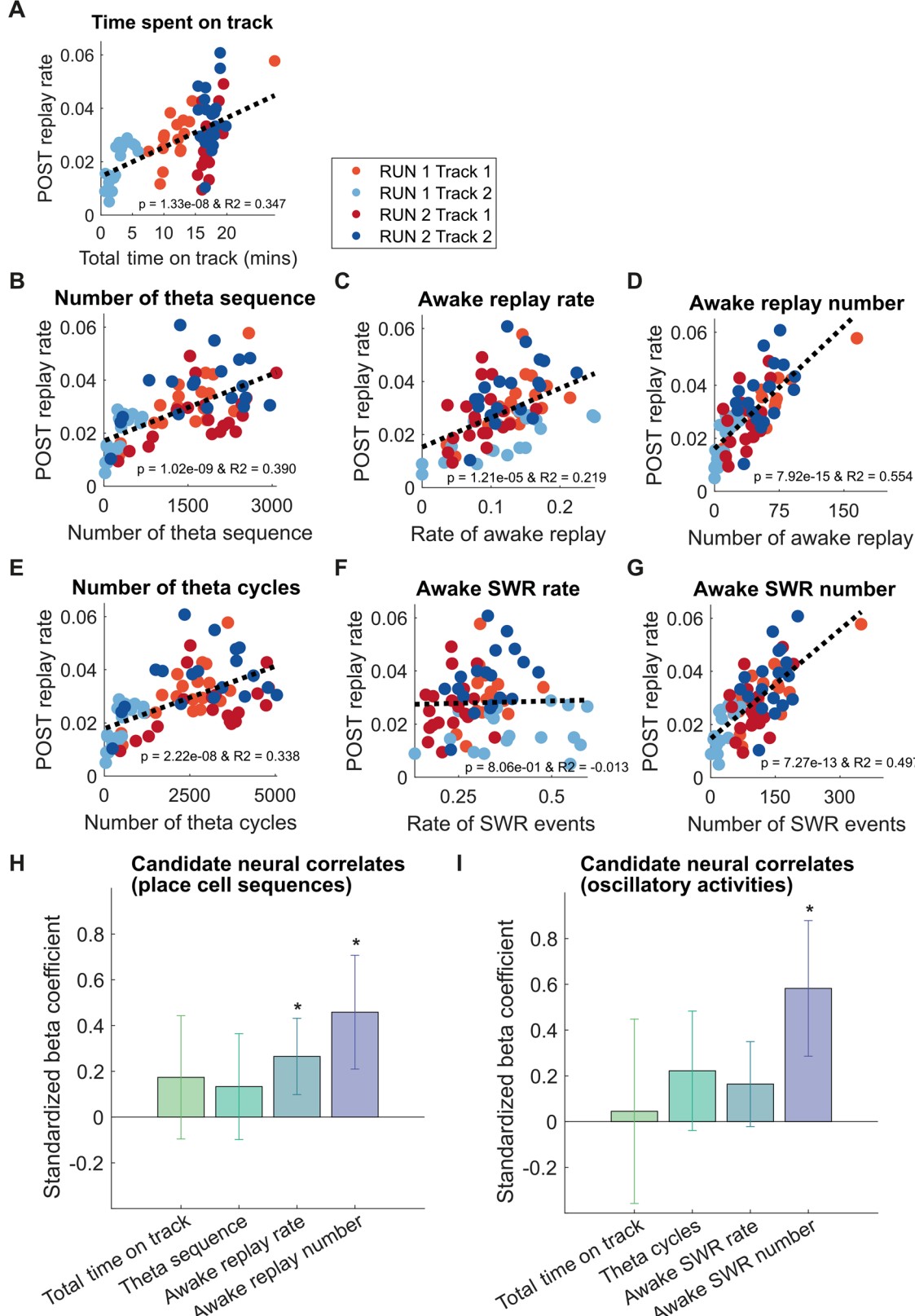

**Fig. 7 | The predictive relationship between the awake replay, theta sequences, and sleep replay. A–G** Linear regression of behavioral or neural metric and rate of POST sleep replay during first 30 min of cumulative sleep. **A** Time spent on track. **B** Number of theta sequence. **C** Awake replay rate. **D** Awake replay number. **E** Number of theta cycles. **F** Awake SWR rate. **G** Awake SWR number. $n = 76$ data points from both tracks during RUN1 and RUN2 of 19 sessions (4 rats). Each data-point is color-coded according to the experimental protocol and track identity

(same as Fig. 1C–F). **H**, **I** Mixed-effect regression for the relationship between candidate neural correlates and sleep replay. **H** place cell sequences. **I** Oscillatory events. Each bar indicates the magnitude of the standardized beta coefficient associated with different factors with the error bar showing the estimated 95% confidence interval. Asterisks (*) where the 95% confidence interval of the standardized beta coefficient does not overlap with 0. See the source data associated with this figure for more information about the models.

from the other track (where a track selective cell only had a place field on one of the tracks). Next, using place cells with place fields on both tracks (a non-overlapping population), we quantified the difference in firing rate and participation between track 1 and track 2 events during PRE, RUN, and POST SWR events. We observed that place cell firing (e.g., firing rate difference or participation track difference) during awake SWR events could explain firing during POST SWR events but not during PRE SWR events (Fig. S9). These results collectively suggest that the relationship between the number of awake the replay and the rate of replay during post-experience sleep is highly conserved for individual neurons, regardless of their magnitude of firing rate on the track or overall level of participation during sleep replay.

## Discussion

Here we have examined two factors that are postulated to influence which memories are prioritized during consolidation- (1) repetition, which we varied by controlling the number of laps run by the rat in a single waking bout, and that when increased resulted in a higher rate of sleep replay during POST1, and (2) familiarity, which varied as a result of the number of laps run by the rat in the previous behavioral episode (RUN1), and when increased resulted in a lower rate of sleep replay in POST2. The effects of prior experience on familiarity may be influenced by the amount of time elapsed between RUN1 and RUN2, and whether or how much time the animal slept during this period, however, these were not varied parametrically in our experiments. We observed that the number of theta sequences increased with the number of laps run by the rat during RUN1, but were unaffected by differences in familiarity during RUN2. In contrast to this, the rate of awake replay was not modulated during RUN1 by the number of laps run by the rat, however, during RUN2, awake replay rate decreased as the prior experience on the track increased (greater familiarity). The change in sleep replay rate for both POST1 (repetition change) and POST2 (familiarity change), while not fully explained by the rate of awake replay or the number of theta sequences, could be most robustly predicted by the cumulative number of awake replay events (Fig. S10). Furthermore, we observed at the level of individual place cells, that the more awake replay events a cell participated in (for a given track), the more likely it would participate in subsequent sleep replay events of that track. Together these data suggest that a memory will be prioritized to be replayed during sleep if there has previously been more awake replay of this memory, cumulatively over the behavioral episode, with the repetition of an experience and the novelty of the context as factors that can generate more awake replay. While the number of theta sequences and local awake replay events tend to be correlated, in general, we would speculate an important distinction- given that theta sequences occur at a much higher number than awake replay events, they provide a more realistic mechanism for the synaptic strengthening across cell assemblies, a likely prerequisite for subsequent sleep replay[17,51,52]. However, once enough theta sequences have occurred for sleep replay to manifest, awake replay likely becomes the more important factor in deciding which memories should be prioritized or triaged during sleep.

Awake replay provides a possible mechanism for explaining how a cue during a behavioral episode can drive online changes that lead to the later prioritization of the memory during sleep replay when this cue is no longer present[53]. Importantly, reward, emotion, and novelty have all been implicated as drivers for an increased rate of awake replay, suggesting that awake replay rates increase with the salience of an experience or the need to update the cognitive map[38,41,48,54,55]. However, even when reward or novelty is lower, longer duration or more repetitive experiences can still lead to more sleep replay and stronger memories[21,56], because while the overall rate of awake replay can be lower, this is compensated by an increased time

period during the behavior when awake replay can occur. It is important to note that our data does not address whether repetition of the spatial trajectory, overall time of the behavioral episode, and/or time inactive during which awake replay normally occurs are key factors modulating sleep replay, as they were largely interdependent in our experimental design. Furthermore, because the animal received a reward for each spatial trajectory, it is difficult to disambiguate between their direct relationship with replay rate. While physical reward can be measured, the subjective value of the spatial trajectory (which may change with novelty and satiety) is challenging to evaluate, and may even change across the behavioral episode. Finally, one key feature of awake replay, not previously observed with theta sequences, is the ability of the hippocampus to reactivate a remote environment while the animal is awake and able to process new cues[40]. Previous work has demonstrated that remote awake replay can be influenced by external cues, such as when an observer rat watches a demonstrator rat running a spatial trajectory, which can trigger the replay of the same trajectory first experienced by the observer rat[57]. Given that human memories can be selectively strengthened during sleep when human subjects are instructed after training that only a subset of the task will be rewarded[22,23], there must be a mechanism by which memories can be tagged post hoc to receive a higher priority for consolidation[58]. We would postulate that when new information is provided after the behavioral task, increasing its future relevance, awake remote replay of the task could theoretically occur, leading to the prioritization of this memory during sleep replay, which would result in a stronger memory post-consolidation. While we cannot discount the possibility that awake replay may also serve other functions, including planning and memory storage[44–47,55], it is also possible that these are simply a byproduct of what part of the cognitive map needs to be tagged to ensure the most efficient prioritization during sleep.

A fundamental question remains - how does awake replay tag a memory, subsequently leading to its prioritization during sleep replay. Which memory replays during sleep at a given moment is likely a stochastic process, however, the probability of a salient memory replaying could still be increased by the brain. We have previously proposed that this bias can be the direct consequence of the increased excitability of cortical neurons that represent the behavioral episode[59]. Recent evidence suggests that the neocortex influences what the hippocampus replays during sleep[8,13,15]. During sleep, epochs of increased cortical activity (up states) precede epochs of increased hippocampal activity (frames) during which replay occurs[8]. Furthermore, the content of cortical activity prior to a sharp-wave ripple can be used to predict what the hippocampus will replay[15]. Similarly, presenting a task-related cue such as a sound, presumably directly driving auditory cortex, will increase the likelihood that the spatial trajectory related to this cue subsequently replays[13]. Together, these data indicate that cortical activity at the start of a cortical upstate influences which memories can replay. Likewise, if cortical circuits representing a memory have an increased excitability during sleep (specifically during the upstate), they may in turn have a more influential vote on what the hippocampus replays, prioritizing this memory for consolidation.

If an increase in excitability within a cortical circuit is required to prioritize a memory for replay during sleep, then how does awake replay play a role in this process? One possibility is that awake replay may occur alongside the reactivation of neuromodulatory pathways (e.g., dopamine), to allow the offline potentiation of co-activated cortical circuits. This appears to be an important distinction between awake and sleep replay, given that the coordinated replay between the VTA and hippocampus has been observed in awake but not sleeping animals[60]. However, this potentiation is likely temporary, as given that the rate of replay decreases over the cumulative sleep (Fig. S4A, B), the potentiating effects of awake replay on sleep replay rate should

diminish with sufficient sleep such that a memory progressively loses its priority for replay. This provides a potential feedback mechanism for the memory triage process where the more a memory replays, the more its replay rate is decreased, effectively de-prioritizing a memory that has sufficiently replayed.

## Methods

The data presented are from four male Lister Hooded rats [Rat 1 (Q-BLU), Rat 2 (P-ORA), Rat 3 (N-BLU), Rat 4 (M-BLU)]. Prior to surgery, rats were housed in pairs and kept at 90% of free-feeding weight with free access to water. The housing room was maintained at a temperature of $22 \pm 2\,°C$, $55 \pm 10\%$ of humidity, and on a 12-h light/dark cycle. All procedures were carried out during the light phase of the cycle in order to facilitate sleep during the rest sessions. All experimental procedures and post-operative care were approved and carried out in accordance with the UK Home Office, subject to the restrictions and provisions contained in the Animals (Scientific Procedures) Act of 1986.

### Behavioral protocol

Prior to the start of recordings, rats were trained for approximately two days, 30 min each, to run back and forth on a linear track with reward delivered at each end. Training occurred in a different room and track from the one used during the recordings. The training period was extended further if required. We designed a 5-day experiment where each day consisted of one recording session where the animal underwent one out of five possible protocols. In each protocol, rats encountered two novel tracks and were allowed to run back and forth for a variable number of laps (with running the entire length of the track back and forth considered a single lap), with a reward consisting of chocolate-flavored milk delivered at each end of the track. The protocols differed in the number of laps the rat had to run on the second track, which was always less than the first track.

A given recording session started with a 1-h sleep period in which the animals were allowed to sleep in the rest box. The rest box consisted of a circular enclosure with walls 50 cm tall and with a towel placed at the bottom, to which the animals had been previously habituated. Next, rats were exposed for the first time to the two novel tracks. These first exposures were separated by 10-min rest in the rest box to facilitate discrimination between the experiences corresponding to each track. In the first track (T1), rats always ran 16 laps back and forth. The number of laps ran on T1 was constant across all five protocols and was used as a control. In the second track (T2), rats ran a lower number of laps, which varied between 1, 2, 3, 4, or 8 laps. The number of laps run on T2 changed every day and the order was pseudo-randomized for each animal. After the first exposure to T1 and T2, the animals were immediately placed back into the rest box and were allowed a 2-h sleep period to consolidate the experience. To ensure that place field stability was achieved for each track, rats were exposed for a second time to the same tracks. This time, animals were allowed to run for 15 min on each track, a sufficient amount of time for acquiring a stable hippocampal representation of the track[61]. As before, both exposures were separated by a 10-min rest in the rest box. As a final step, rats were allowed to sleep for another hour.

### Experimental setup

A key element of the protocol was to ensure the animals encountered a new environment and two novel tracks every day. With that goal in mind, we designed a modular maze consisting of wooden planks of medium-density fiberboard (MDF) cut to different sizes. Pieces of MDF could be attached together into different configurations using wooden dowels, thus easily creating multiple track shapes using the same pieces. All rats were exposed to the same track shapes, with the order pseudo-randomized across sessions and tracks. This was done to minimize any potential effect of a particular track shape on the results

of the experiment. To facilitate tracking and prevent slipping, the MDF pieces were painted with gripping black paint (Blackfriar, UK) and sprayed with matt black paint (Hycote, UK). To further ensure hippocampal remapping across days and tracks (i.e., to ensure tracks were different enough to be recognized as such), we used a variety of textures to cover the tracks and also made use of distal visual cues hanged on the walls to simulate different rooms. Additionally, large vertical polypropylene black sheets were used as panels to divide the room, thus creating the illusion of sub-rooms and changing the overall spatial configuration of the room across days. The whole recording setting was surrounded by black curtains mounted on the ceiling and lit up with dimmable blue LED strips (CPC, 12 V).

The behavioral task was automated using custom-made software with Bonsai (https://bonsai-rx.org//) and Arduino board (http://www.arduino.cc/). The software monitored the position of the animal through four webcams placed in the ceiling (Logitech C930E 1080p HD Webcam) and controlled the delivery of the liquid reward by activating the infusion pumps (dual Aladdin, WPI) upon the animal's entry in the region of interest set near the reward wells, at each end of the track.

### Electrophysiological recordings

A large-scale independently movable microdrive array was used to record single units[62]. The design of the microdrive was modified to both improve flexibility and reduce weight, to increase the number of tetrodes, and to duplicate the number of targeted areas. Rat 2 (P-ORA) was implanted with a dual-hippocampal microdrive, while the three remaining were implanted with a microdrive targeting both the dorsal hippocampus and primary visual cortex (V1).

The body of both types of microdrive was designed using an online 3D modeling software (Vectary Inc.) and later 3D-printed (Form2 3D printer, Formlabs). Each microdrive contained 24 independently movable tetrodes carried inside two polyimide tubes: an inner (ID: 0.0035"; OD: 0.0055", IWG) and outer (ID: 0.0071"; OD: 0.0116", IWG). The outer polyimide contained the inner one, and it was used as a guide tube to direct the inner polyimide through the body of the microdrive. The inner polyimide was glued to both the tetrode and the movable screw, allowing the tetrode to slide up and down while protecting it from bending during the movement. The dual-hippocampal microdrive had two outputs with 12 tetrodes each that targeted the hippocampal region in each hemisphere. For the hippocampal-visual cortex microdrive, 16 tetrodes were used to record from the hippocampal area, while the remaining 8 tetrodes targeted the visual cortex. Tetrodes were assembled using four twisted tungsten microwires (12 μm diameter, Tungsten 99.95% CS, CFW), individually gold-plated to <200 kΩ impedance (NanoZ, White Matter LLC). To protect the microdrive and achieve a better grounding, we designed a cone that could both contain the microdrive and act as a Faraday cage. The structure of the cone was built from aluminum foil glued into a plastic sheet, and grounded to the Electrode Interface Board (EIB) of the microdrive through a soldered wire.

### Surgical procedure

For the surgical implantation of the microdrive, rats were induced and maintained under isoflurane anesthesia (1.5–3% at 2 L/min). Carprofen (0.1 mL/100 g animal weight in a solution of 1:10, Pfizer Ltd, UK) and Baytril (10 mg/Kg, Bayer) were given pre-surgically to prevent pain and infection. Following the induction of anesthesia, the animals were shaved and placed on the stereotaxic frame with ear bars. After disinfecting the skin with antiseptic (Betadine) and saline, an incision was made to expose the skull, which was carefully cleaned with 10% hydrogen peroxide diluted in phosphate-buffered saline and a phosphoric acid-based etching gel agent (37.5%, Gel Etchant) to improve bonding to dental acrylic. Throughout the surgery the animal's body temperature was kept constant with a heating pad.

For rats implanted with the hippocampal-visual cortex microdrive, the craniotomy aimed at the pyramidal cell layer in the CA1 area of the right dorsal hippocampus, targeted the coordinates (from bregma: ML = 2.5 mm, AP = 3.72 mm). For the rat implanted with the double-hippocampal microdrive, bilateral craniotomies were made above the dorsal hippocampal CA1 cell layer with coordinates (from bregma: ML = +/−2.5 mm, AP = −3.72). Two small metal screws with a soldered wire were placed in the frontal right parietal bone and the right occipital bone (above the cerebellum), where they served as reference and ground, respectively. Extra screws were added in strategic positions in the skull to act as an anchor for the implant. The microdrive and screws were fixed in place using metabond (Super Bond C&B) and dental acrylic (Simplex Rapid ®, Kemdent, UK). Finally, the skin was sutured and the animal was left to recover in a heated chamber and monitored until fully recovered all motor functions and the ability to drink and eat. As post-surgical care, rats were administered with low doses of analgesics (Metacam, 1.5 mg/mL Oral Suspension for Dogs 10 mL, Boehringer Ingelheim) during 72 h. Animals were housed individually and allowed to recover with food and water ad libitum for a week, before returning to being kept at 90% of their free-feeding weight.

After recovery, implanted rats were screened for hippocampal single units by gradually lowering the tetrodes until reaching the CA1 pyramidal cell layer. Neuronal activity and position data were acquired with a 96-channel digital acquisition system (Neuralynx, DigitalLynx). The signals were pre-amplified and digitized on the head-stage at a sampling rate of 30 kHz. Local field potential and spikes were then band-pass filtered in the Neuralynx acquisition unit at 0.1 Hz and 600 Hz–6000 Hz, respectively. Video tracking was acquired at 25 fps using a camera also connected to the Neuralynx acquisition device, while two LEDs mounted in the head-stage were used to infer head-direction and position of the animal. Once tetrodes reached CA1, recording sessions started.

Upon completion of the experiments, animals were deeply anaesthetized with isoflurane (3% isoflurane with an oxygen flow rate of 2 L/min) and the locations of the recording sites were marked with electrolytic lesions by passing current through an electrode of each tetrode (10 s, 30 μA). Animals were then terminated via an intraperitoneal injection of a lethal dose of Euthatal (0.5 mL/100 g, sodium pentobarbital) and perfused transcardially with saline (0.9% sodium chloride solution) followed by 10% Formalin. The fixed brains were then removed from the skull while carefully extracting the implanted microdrive, and post-fixed in 10% Formalin at 4 °C for a minimum of 48 h. Next, the brains were placed in a container with 30% sucrose solution to achieve cryoprotection. Once the brains sunk in the container, they were mounted on a block with Optimum Cutting Temperature (OCT) and sectioned coronally with a cryostat (Leica, CM1850 UV) at a thickness of 30 μm. The obtained brain slices were then wet-mounted onto superfrost plus slides (Thermo Scientific), Nissl-stained, and coverslipped using DPX mounting media (Sigma Aldrich). Finally, slices were examined under a Leica DMi8 microscope in order to detect the tetrodes track reaching into the hippocampal CA1 pyramidal layer. Whole-slice images were obtained using bright-field settings at 25X magnification.

## Spike sorting and unit isolation

Spiking data was sorted using the semi-automatic clustering software KlustaKwik 2.0 (K.Harris, http://klustakwik.sourceforge.net/) and then manually curated with Phy-GUI (https://github.com/kwikteam/phy). Putative single units were isolated based on the spike waveform, auto-correlograms, and their stability across the recording session. The rest of the clustered activity was classified in either multi-unit activity or noise.

Position data was collected by tracking the LEDs attached to the head-stage during recordings. The instantaneous speed was calculated as the derivative of the position data. Tracking errors due to large reflections or transient failure to detect the head-stage's LED were cleaned offline by (1) constraining tracking areas around the track's and rest box's area; (2) removing large jumps between consecutive pixels by setting a maximum distance jump of 40 cm; (3) setting a speed threshold of 100 cm/s. All discarded position points and their corresponding timestamps were then linearly interpolated. Cleaned tracking and speed data were next converted from pixels to cm/s and linearized from a two-dimensional coordinate into a single coordinate ($x$, the distance traveled along the track).

## Local field potential analysis

The power spectral density (PSD) of the hippocampal LFP was quantified using Welch's method (pwelch, MATLAB) with 2 s windows with 50% overlap for the entire session. The PSD was used to identify the channels with higher power for theta (4–12 Hz) and ripple (125–300 Hz) oscillations, as well as the channel with the largest difference in normalized theta to ripple power. The LFP of the selected channels was down-sampled from 30 kHz to 1 kHz and band-passed filtered in forward and reverse directions in order to avoid phase delays (MATLAB command filtfilt). The instantaneous phases were estimated using the Hilbert transform.

## Putative sleep quantification

Putative sleep was defined by periods of immobility (windows of 60 s with velocity lower than 4 cm/s) accompanied by transient periods of high multi-unit activity ($z$-score greater than 0). In order to reduce noise levels when setting the multi-unit activity (MUA) threshold, only the most active units (top 1/3 units in terms of total spike counts) were used. To account for intrasubject variability, both the velocity and MUA threshold were visually checked for each data and session and corrected if needed. As a control, we compared our putative sleep detection method with one based on the LFP activity, as it has been commonly used in other studies[34,63]. For each session, we calculated the LFP theta/delta ratio based on the channels with higher theta and delta power, respectively. Awake periods were classified as high $z$-scored theta power (>0.5) and high velocity (>4 cm/s). NREM periods were characterized by low $z$-scored theta power (<0.5), low mobility (<4 cm/s) and high $z$-scored theta/delta ratio (>0.5), while REM sleep periods were classified as high $z$-scored theta power (>0.5), low mobility (<4 cm/s) and high $z$-scored theta/delta ratio (>0.5). For the purposes of this analysis, NREM and REM periods were merged, and then compared to the putative sleep periods obtained with our detection method. The putative sleep detected periods with each method were highly correlated (mean all sessions = 0.51 ± 0.15, with a mean $p$-value < .001 ± 0.002).

## Place cell classification

Putative principal hippocampal cells were identified by selecting units with a half-width half max (HWHM) larger than 500 μs and mean firing rate <5 Hz across the entire recording session. For place cell classification, spike trains were speed-filtered to only include the spiking activity between 4 cm/s and 50 cm/s. A principal cell was classified as a spatially selective place cell if it had a minimum peak firing rate that was >1 Hz in its unsmoothed ratemap for at least one of the two linear tracks.

To generate firing ratemaps (the spike histogram divided by the total dwell time at each position bin), the position data was discretized in 2 cm bins for visualization and plotting, and 10 cm bins for Bayesian decoding. Only raw (unsmoothed) ratemaps were used for all Bayesian decoding analyses.

## Bayesian decoding of animal's spatial trajectory

A naïve Bayesian decoding algorithm was applied to reconstruct the estimated position of the animal during behavior and replay events

based CA1 hippocampal spiking activity[64]:

$$P(x|n) = CP(x)\left(\prod_{i=1}^{N} f_i(x)^{n_i}\right) exp\left(-\tau \sum_{i=1}^{N} f_i(x)\right) \quad (1)$$

where $P(x|n)$ is the probability of the animal being at a specific position given the observed spiking activity, $C$ is a normalization constant, $x$ is the animal's position, $f_i(x)$ is the firing rate of the $i$th place field at a given location $x$, and $n$ is the number of spikes in the time window $\tau$. The normalization constant was defined as the summed posterior probabilities across both tracks for first exposure and second exposure separately. We used non-overlapping temporal windows of 250 ms to decode the animal's location while running on the track, 20 ms windows to decode replay events, and 10 ms windows to decode theta sequences.

The decoding error was defined as the difference between the real location of the animal and the estimated position with maximum likelihood. The decoding accuracy of each session was analyzed by quantifying the confusion matrices of the median decoding error. Sessions with a decoding error higher than 15 cm in the re-exposures were discarded.

### Detection of replay event

Replay trajectories were decoded using a Bayesian decoding algorithm, as described in Tirole et al. (2022)[27]. Detection of candidate sharp-wave ripple (SWR) associated replay events was based on the thresholds set on both multi-unit activity (MUA) and ripple-band power. MUA was first binned into 1 ms steps and smoothed with a Gaussian Kernel (sigma = 5 ms). Only MUA bursts with a maximum duration of 300 ms and $z$-scored activity over 3 were included. Next, the ripple-band filtered LFP signal was smoothed with a 0.1 s moving average filter, and calculated the amplitude of ripple-band filtered signal using the Hilbert transform. The candidate replay event was required to pass ripple threshold set at $z$-score of 3. Candidate replay events passing both thresholds were next speed-filtered (above 5 cm/s), and discarded if the events involved less than 5 different units active or if their duration was below 100 ms or above 750 ms. Therefore, each event should contain at least five 20 ms time bins for decoding. Events detected within 50 ms of each other were combined.

Replay events were classified as rest, sleep, or awake local replay. Local awake replay was defined as replay events where its decoded track identity matched the track on which the animal was currently on. Replay events during POST1 and POST2 were classified as sleep replay if they occurred when animals' mean moving speed within a 1-min time bin was lower than 4 cm/s accompanied by transient periods of high multi-unit activity ($z$-score greater than 0), otherwise they were classified as rest replay. For some analyses that examined the relationship between awake SWR events and sleep replay, candidate SWR events before decoding were used. To classify a given candidate replay event as significant for a given track, we quantified the weighted correlation of the posterior probability matrix, which calculates the correlation coefficient between time ($T$) and decoded position ($P$) by weighting each estimated position by its decoded probability ($prob$):

Weighted mean:

$$m(x; prob) = \frac{\sum_{i=1}^{M}\sum_{j=1}^{N} prob_{ij} x_i}{\sum_{i=1}^{M}\sum_{j=1}^{N} prob_{ij}} \quad (2)$$

Weighted covariance:

$$cov(x, t; prob) = \frac{\sum_{i=1}^{M}\sum_{j=1}^{N} prob_{ij}(x_i - m(x; prob))(t_j - m(y; prob))}{\sum_{i=1}^{M}\sum_{j=1}^{N} prob_{ij}} \quad (3)$$

Weighted correlation:

$$corr(x, t; prob) = \frac{cov(x, t; prob)}{\sqrt{cov(x, x; prob)cov(t, t; prob)}} \quad (4)$$

Where $x_i$ is the $i$th position bin, $t_j$ is the $j$th time bin and $prob_{ij}$ is the probability at the position bin $i$ and time bin $j$.

To determine the statistical significance of the replay events, we compared each candidate event's weighted correlation score to three different shuffled distributions:

1. Spike train circular shift, in which the spike count vectors for each cell were independently circularly shifted in time within each replay event, prior to decoding.
2. Place field shift, in which each ratemap was circularly shifted in space by a random amount of position bins prior to decoding.
3. Circular shift of position, in which posterior probability vectors for each time bin were independently circularly shifted by a random amount.

If the score of the candidate event was greater than the 95th percentile of the distribution for all three shuffles then the event was considered to be significant. In a few occasions, replay events were found to be significant for both tracks (hereafter referred as multi-tracks events). The mean proportion of multi-tracks events during first exposure and re-exposure were $0.0431 \pm 0.0048$ and $0.0571 \pm 0.0047$, respectively. Those events were assigned to one of the tracks by computing the Bayesian bias score for each track. Each score was calculated as the sum of the posterior probability matrix for one track, normalized by the total sum across tracks. To assign the replay event to a specific track, the Bayesian bias score was required to be greater than 60%, otherwise the event was discarded. We do not think the proportion of multi-track events is high enough to have any major impacts on our findings.

To optimize the detection of replay events and avoid discarding a minority of events due to noisy decoded probability at the beginning or the end of the event, candidate replay events were split into two segments where the midpoint was determined based on the minimum MUA activity in the middle third of the candidate event. Halved segments were decoded and analyzed for statistical significance independently if they passed the same selection criteria described above with the exception of the $p$-value threshold adjusted to $p < 0.025$ to account for multiple comparisons.

Sleep replay activity was analyzed over the first 30 min of cumulative sleep (in all figures unless explicitly indicated), to allow a balanced comparison between POST1 and POST2 sleep. To calculate the average rate of sleep replay, we divided the number of replay events over 30 min of cumulative sleep by 30 min. For local awake replay, we first counted the total number of local replay events occurring on the linear track and divided the number of events by the total amount of time spent immobile (period when speed <4 cm/s) to calculate the average rate of local awake replay.

### Detection of theta sequences

Theta cycles were detected using a peak-trough detection method. A sliding window was used to search for extrema within a voltage range, which was set to be 25% of the median amplitude of the total signal across the session (code based on online script http://nocurve. com/virtual-lab/finding-peaks-and-troughs-in-a-noisy-curve/). Consecutive peaks and troughs were deleted by selecting the maximum extrema. Theta cycles were speed-filtered (below 5 cm/s) and excluded if they occurred in the reward zone (within 20 cm from each end of the track). Cycles shorter than 80 ms or longer than 200 ms were discarded. Theta phase was calculated extracting the angle from the Hilbert transform and binned into windows of $2\pi$ length (equivalent to a complete cycle). Next, spike phases were extracted through

linear interpolation and assigned to each window. Theta windows with less than 2 active units were excluded. Phase bin edges from each window were then interpolated back to time and spike sequences contained within each time window were decoded using a naïve Bayesian decoder (described above). Theta sequences were decoded for each running direction, using directional ratemaps, and repeated on a lap-by-lap basis using the smoothed ratemaps from the corresponding lap. The resulting posterior probability matrix for each sequence was then centered on the animal's current location (±40 cm), such that the actual position of the rat in the window was at 0 cm. All decoded sequences for one running direction were next reversed in order to average all the posterior probability matrices across all theta cycles.

Similar to replay detection, theta sequence was quantified using weighted correlation. The significance of theta sequences was determined by comparing the obtained score to three different shuffled distributions:

(1) Spike train circular shift, a pre-decoding shuffle that alters the spike timing of each cell within theta sequence by circularly shifting the spike times in the temporal dimension;

(2) Circular shift of position, which circularly shifts the estimated positions within a time bin,

(3) Circular shift of decoded theta phase, which disrupts the phase domain by circularly shifting the estimated phase in each position bin, but maintains the relationship between position and spike probability.

If the score of the candidate theta sequence event was greater than the 95th percentile of the distribution for all three shuffles then the event was considered to be significant.

### Simple and mixed-effect linear regression analysis

Simple linear regression was used to quantify the predictive relationship between the metric of interest and the rate of subsequent sleep replay for track 1 and track 2 events across both exposure sessions (MATLAB function *fitlm*). To further examine the relative magnitude of the effects of different neural and behavioral variables to the rate of sleep replay, we applied mixed-effect linear regression analysis (MATLAB function *fitlme*). For the effect of awake replay and theta sequence on sleep replay, we fitted a mixed-effect model for sleep replay rate with the rate and number of local awake replay, the number of theta sequence, and the total time spent on the track as fixed effects and animal identity as a random effect:

Sleep replay rate ~ rate of awake replay + number of awake replay + number of theta sequence + total time spent on the track + (1/animal).

For the effect of awake SWR and theta cycles on sleep replay, we fitted a mixed-effect model for sleep replay rate with the rate and number of local awake SWR events, the number of theta cycles, and total time spent on the track as fixed effects and animal identity as random effect:

Sleep replay rate ~ rate of awake SWR events + number of awake SWR events + number of theta cycles + total time spent on the track + (1/animal).

All metrics were converted into standardized z-scores such that the standardized beta coefficient can be used to compare the relative weight of each fixed-effect term. We calculated the 95% confidence interval for the standardized beta coefficient such that the relative predictive power of the metric would be considered statistically insignificant when it overlapped with zero.

### Reporting summary

Further information on research design is available in the Nature Portfolio Reporting Summary linked to this article.

## Data availability

Data availability via a public repository is delayed due to this dataset's use in several manuscripts in preparation. However, all data are available immediately upon reasonable request to the corresponding author. Source data are provided in this paper.

## Code availability

All custom-written MATLAB code is available on *Zenodo*: https://doi.org/10.5281/zenodo.10085294.

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

## Acknowledgements

We thank Margot Tirole, Sophie Renaudineau, Lilia Kukovska, Julieta Campi, and Joanna Holeniewska for technical assistance; members of the Bendor Lab for valuable discussion; and Soraya Dunn and Tom Wills for their comments on the manuscript. This work was supported by the European Research Council starter grant (CHIME) (D.B.), the Human Frontiers Science Program Young Investigator Award (RGY0067/2016) (D.B.), the Biotechnology and Biological Sciences Research Council Research grant (BB/T005475/1) (D.B.) and the Medical Research Council scholarship (MR/N013867/1) (M.T.). The Titan Xp used for this research was donated by the NVIDIA Corporation.

## Author contributions

M.H.G. and D.B. designed the experiment, M.H.G. collected the data, M.H.G. contributed to methodological development, M.H.G. and M.T. analyzed the data, D.B., M.H.G., and M.T. wrote the initial draft, M.T. and D.B. revised the manuscript.

## Competing interests

The authors declare no competing interests.

## Additional information

**Publisher's note** Springer Nature remains neutral with regards to jurisdictional claims in published maps and institutional affiliations.

