## [Peer Review File · Nature Communications]

The Role of Experience in Prioritizing Hippocampal ReplayREVIEWER COMMENTS

Reviewer #1 (Remarks to the Author):

Gorriz et al. examine in vivo recordings of place cells while rats explore two environments with differing levels of exposure. After a rest period, the rats are re-exposed to the tracks, this time with similar levels of exposure. The authors report that during the first rest period, replay events are biased toward the more-repeated track, while during the second rest period, replay events are biased toward the less familiar track. The post-sleep replay bias is more strongly associated with a bias in awake, on-task replay than with on-task theta sequences, arguing that one role of awake replay is to facilitate or prime the network to replay salient experiences during sleep-based replay for memory consolidation.

In general, I find this to be a well-organized, clearly described study. The results are novel and important, speaking to network-level mechanisms of hippocampal memory formation/consolidation, and I find the conclusions convincing. I have only a few relatively minor criticisms and analysis requests to clarify interpretation of the authors' core findings.

1. The vast majority of the authors' findings are dependent upon their ability to faithfully decode the information content of SPW/Rs and theta sequences to identify events which encode coherent information and those which do not. The authors do show low decoding errors in Sup Fig S2, and I have strong confidence that when the authors identify a SPW/R that encodes a sequence, that is very likely a replay. However, the quantification commonly used by the authors is the *rate* of replay, which critically requires the authors to be able to accurately distinguish replay-encoding SPW/Rs from SPW/Rs encoding 'fragmented' information. Importantly, this categorization is heavily dependent upon a number of factors, including total numbers of simultaneously recorded neurons, quality of place fields, etc.

One concern I have regarding the authors' ability to decode accurately is in the rate of replay in POST1 for Track2 when the number of laps may be insufficient to properly calculate place fields given large variability from lap-to-lap (Olypher & Fenton, *Neuroscience*, 2002). Indeed, the decoding error seems considerably larger for Track2/RUN1 with fewer laps (Sup Fig S2). The authors attempt to account for this concern in Sup Fig S5 by decoding POST1 with place fields from RUN2, but given the possibility of remapping, this may not be a fair comparison. Rather, it seems more appropriate to compare Track1 and Track2 replay using place fields calculated from equivalent numbers of laps on RUN1. Thus, I would like to see Sup Fig S5 expanded to include decoding of POST1 SPW/Rs using place fields calculated on only the first X number of laps for both Track1 and Track2 (where X is the number of laps for Track2).

2. Related to the first point, the data in Figure S7 are important to provide a decoding-independent, single-cell-based confirmation of the authors' main finding. However, while the data in Fig S7 argue that firing in awake replay is a strong determinant of firing in post-sleep replay, these analyses have two caveats. First, they do not account for overall firing rates of these cells in the two tracks. As it is known that firing rates are correlated across states (Buzsaki & Mizuseki, *Nat Rev Neurosci*, 2014), it is possible that differences in firing rates during theta sequences between track 1 and 2 account for differences in firing rates in awake replay and post-experience sleep replay. In fact, I would imagine plotting the x-axes for Sup Fig S7 for awake theta sequence difference rather than awake replay difference would produce the same correlations (as suggested by the data in Fig 7B). Second, these analyses still require the authors to faithfully identify replay-encoding SPW/Rs, which weakens the argument that these are decoding-independent measures.

To address these concerns, I would like the authors to repeat the analyses for Figure S7, but correlate total firing in the task to total firing across all pre-experience SPW/Rs or to total firing across all POST SPW/Rs to rule out the possibility that the observed correlations are simply due to consistent, inherent

differences in activity from cell to cell.

Alternatively (or perhaps in addition to the above suggestion), the authors could examine the activity of neurons with place fields in **ONLY** Track1 or Track2. The authors could compare these "track-selective" place cells during run and during POST rest. Assuming that Track1-selective cells have similar place field properties as Track2-selective cells (equivalent place field size, information/spike, firing rate, etc.), the authors should be able to predict differences in the firing of Track1- vs. Track2-selective neurons in POST SPW/Rs that are not trivially explained by on-task firing rates or decoding accuracy.

3. In the discussion, I think it is important to expand on the notion of familiarity vs. repetition. It seems to me that familiarity is just repetition with a larger temporal gap between experiences. Do the authors think that repetition is the number of similar experiences during a single waking bout, while familiarity is comparing experiences on opposite ends of sleep/rest periods? How much time do the authors think defines "repetition" vs. "familiar"? Given the opposite network effect of these two conceptually similar terms, it is worth expanding on this notion in a paragraph or so. For example, are there known molecular/synaptic/circuit changes that might 'tag' temporal events for a short time, or do the authors think that sleep or sleep replay 'resets' the repetition counter? If experiences have simultaneously different levels of familiarity AND repetition, do the authors think one or the other would be dominant? I realize the discussion would be somewhat speculative in nature, so I don't expect the authors to go into great detail (indeed, the discussion could be shorter than this paragraph I am writing to explain the matter to be discussed!), but I think this is important to place the work into a broader context.

4. In Fig 1 and Sup Fig 1, the authors quantify that total moving time, immobility, and moving speed are equivalent between tracks 1 and 2 in RUN2. However, the important behavioral measure here is likely total experience, which is more directly quantified by number of laps or total distance covered (particularly important since in RUN2 the rats get a fixed amount of time rather than a fixed number of laps). Given the similarity in velocity and time spent mobile, it is very likely that total distance covered and total number of laps will be similar between tracks 1 and 2 in RUN2, but I believe this is important to quantify in order to make this point clear and facilitate proper interpretation of the core finding.

5. The definition of sleep is a little unclear. If the rat moved less than 4 cm/s for a window of 60s, was it assumed to be asleep for the entire window? It seems more likely that rats will be immobile but still awake for a short time before falling asleep, and therefore it may be better to exclude some duration (perhaps the first 30-60 s) of immobility from sleep quantification. Please clarify this in the methods. In addition, given the absence of EMG measurements, the authors should refer to this as putative sleep.

6. Minor Issues:

The figure legend for Fig S4 (C,D) says "Cumulative sleep replay", but I believe that should be "Cumulative rest replay."

In the Methods, please state the total number or percentage of the "few occurrences" in which replay events were significant for both tracks to allow assessment of the accuracy/precision of the decoding.

The authors restrict MUA activity analyses to "only the most active units." For purposes of replication, please describe the criteria used for selection here.

In the final sentence of the abstract, I believe that "selectively" should be "selective," or perhaps the "the" should be removed.

In Sup Fig S3, should the regression b value in panel C be negative?

Reviewer #2 (Remarks to the Author):

This Gorriz et al. manuscript sets out to address which memories are prioritized for sleep- or rest-dependent consolidation, as inferred from replay of dorsal CA1 place cell sequences in rats. The central protocol aims to vary familiarity with linear tracks by allowing 4 rats to run back and forward for different numbers of trials. The rats are rewarded at each end of the tracks, but are not required to learn – the assumption is that more experience will translate to more en passant learning. The authors therefore hypothesize that more trials should lead to more familiarity, and that re-exposure to a familiar track should demand less mnemonic processing and induce less subsequent replay. Replay is quantified from place cell patterns using Bayesian decoding of location.

The basic design is reasonable and the recording/replay detection methods used are standard for the field. As the authors state, their results “support previous work reporting that more reactivations occur during behavioral episodes for novel environments compared to familiar ones (Cheng and Frank, 2008; O’Neill et al., 2008; Hwaun and Colgin, 2019)”. However, as the manuscript stands, my sense is that some of the main findings have alternate interpretations and I would appreciate clarification of the following:

1. My main concern is that the protocol covaries # trials and # rewards: because of increased runs on T1, the rats receive more reward on T1. Could all the results be explained by this, meaning the findings are similar to those of Singer & Frank (<https://pubmed.ncbi.nlm.nih.gov/20064396/>)? I do not see how repetition and reward can be disentangled using this paradigm.
2. The behavioral analyses indicate that rats pause more and move more on T1 than T2 during RUN1, but that running behavior on T1 and T2 is similar during RUN2. Since the rats do not show discrimination between T1 and T2 on RUN2 re-exposure, one interpretation is that the increased prior exposure to T1 has no significant bearing on their behavior, i.e. their memory. I certainly think that extrapolating to “perceived future relevance” is a stretch.
3. What is the nature of the running and pausing behaviors (particularly on T1 during RUN1), where on the track do awake replay events occur and which portions of the track are replayed? These details may help to illuminate whether/how familiarity/reward shape coding of the environment under these conditions.
4. How large are “large ensembles” and did the # or proportion of units participating in replay events vary systematically with conditions?
5. How much sleep and rest was there during POST1 and POST2? Was the 30min of cumulative sleep selected for analysis distributed similarly across time, independent of session type?
6. Given such similar results for POST “rest” replay (Fig S6) and “sleep” replay, does sleep matter? It might, in particular for hippocampal-neocortical interactions – but these are not interrogated here.
7. The final 3 paragraphs of the Conclusions are speculatively related to this study and could be shortened – particularly the sections on cortical interrelationships with hippocampal replay.

Reviewer #3 (Remarks to the Author):

This manuscript by Gorriz et al describes their study on how the sleep replay of place cell activity patterns representing a space was influenced by the animal’s experience in the space. The authors specifically targeted two aspects of the experience: repetition and familiarity. They found that more repeated running on a novel track led to more sleep replay and same level of running on a familiar

track reduced sleep replay. The authors also examined how the sleep replay was related to the awake replay or the theta sequences on the same track and found that the amount of sleep replay after all types of experience was best correlated with the accumulative number of awake replay events. The authors conclude that sleep replay for memory consolidation is prioritized toward repeated experience in a novel environment.

I found the study well-designed, the data analysis straightforward and clean, and the manuscript well written. The conclusion is convincing and well-supported by the data presented. Although replay of hippocampal place cells is extensively studied, how it is influenced by the nature of prior experience was previously studied only in separate experiments. This manuscript directly compares sleep replay in novel/familiar environments with different number of running laps, all in same animals. I do not have serious concerns on the data or the conclusion. I only have a couple of suggestions for the authors to consider.

1) The authors used number of replay events and rate of replay to quantify sleep (and awake) replay. One suggestion is that they may consider using z-score-normalized number of replay events relative to its cell-identity shuffled distribution. Due to differences in templates or number of candidate events across animals/sessions, averaging among animals or comparing different conditions may not be accurate. Using normalized number of replay events should improve this aspect.

2) The authors considered theta sequences and awake replay as potential mechanisms for prioritizing sleep replay. Although they are correlated in some aspects, they are all phenomena of place cell patterns. There is no direct evidence for a mechanistic link among them. I suggest not to present one as a mechanism of another, only to examine which is best explained or correlated with which.

We would like to thank the editors and the three reviewers for the time and effort taken in reviewing our manuscript and providing constructive feedback.

Below, we address questions and comments raised by the reviewers, including new analyses and additional supplemental figures in the revised manuscript. Our response to the reviewers' comments follows the format in which the points from the reviewer are in *black*, while our responses are in *blue*.

Additionally, we have updated the manuscript based on the guidance on sex and gender reporting. In particular, we now also indicated the fact that all rats used in our study were male in the abstract.

Reviewers' comments

Reviewer #1 (Remarks to the author):

1. The vast majority of the authors' findings are dependent upon their ability to faithfully decode the information content of SPW/Rs and theta sequences to identify events which encode coherent information and those which do not. The authors do show low decoding errors in Sup Fig S2, and I have strong confidence that when the authors identify a SPW/R that encodes a sequence, that is very likely a replay. However, the quantification commonly used by the authors is the **rate** of replay, which critically requires the authors to be able to accurately distinguish replay-encoding SPW/Rs from SPW/Rs encoding 'fragmented' information. Importantly, this categorization is heavily dependent upon a number of factors, including total numbers of simultaneously recorded neurons, quality of place fields, etc. One concern I have regarding the authors' ability to decode accurately is in the rate of replay in POST1 for Track2 when the number of laps may be insufficient to properly calculate place fields given large variability from lap-to-lap (Olypher & Fenton, Neuroscience, 2002). Indeed, the decoding error seems considerably larger for Track2/RUN1 with fewer laps (Sup Fig S2). The authors attempt to account for this concern in Sup Fig S5 by decoding POST1 with place fields from RUN2, but given the possibility of remapping, this may not be a fair comparison. Rather, it seems more appropriate to compare Track1 and Track2 replay using place fields calculated from equivalent numbers of laps on RUN1. Thus, I would like to see Sup Fig S5 expanded to include decoding of POST1 SPW/Rs using place fields calculated on only the first X number of laps for both Track1 and Track2 (where X is the number of laps for Track2).

The reviewer makes a great point that our finding about difference in track 1 and track 2 replay depends on the ability to decode sequential pattern during SWR against track 1 and track 2 spatial templates. This means that low level of track 2 replay during first exposure can be due to poor sampling of place fields. Therefore, we agree with the reviewer that it is crucial to address this concern by comparing track 1 and track 2 replay using place fields calculated from equivalent numbers of laps on RUN1. To make the results more comparable across different conditions, we compared replay using place fields calculated from the final single lap for both track 1 and track 2 regardless of the total laps ran. We believe this would ensure the amount of neural data contributing to the place field template is equivalent.

Consistent with our main findings, we still observed a significantly higher sleep replay rate in POST1 for track 1 compared to track 2 across protocols (**two-tailed Wilcoxon signed rank test $p = 0.007$, n**

= 19). We have now included this analysis into **Fig S6** and have updated the manuscript on page 7 accordingly:

*“As an alternative approach, we measured sleep replay during POST1 using place fields obtained in RUN2, which were fully stabilized after 15 min of running on each track. Using this approach, we observed the identical trend of a higher rate of track 1 sleep replay (relative to track 2) during POST1 (**Fig S6A**). Given the possibility that the place cell representations may partially remap between RUN1 and RUN2, we also quantified sleep replay during POST1 using place fields calculated from the final (single) lap for both track 1 and track 2 regardless of the total laps ran. The result remained consistent with the main finding (**Fig S6B**).”*

Figure S6. Sleep replay rate track difference during POST1 remained significantly different when alternative place field templates were used for replay decoding. Rate of sleep replay for track 1 and track 2 during first 30 mins of cumulative sleep of POST1, after detecting replay events decoded using place fields from **(A)** RUN2 and **(B)** the final lap from RUN1. The lap number associated with each track during RUN1 was made in bold. Each data point represents the mean sleep replay rate within a session, color-coded according to the experimental protocol (** $p < 0.01$, *** $p < 0.001$, two-tailed Wilcoxon signed rank test)

2. Related to the first point, the data in Figure S7 are important to provide a decoding-independent, single-cell-based confirmation of the authors' main finding. However, while the data in Fig S7 argue that firing in awake replay is a strong determinant of firing in post-sleep replay, these analyses have two caveats. First, they do not account for overall firing rates of these cells in the two tracks. As it is known that firing rates are correlated across states (Buzsaki & Mizuseki, Nat Rev Neurosci, 2014), it is possible that differences in firing rates during theta sequences between track 1 and 2 account for differences in firing rates in awake replay and post-experience sleep replay. In fact, I would imagine plotting the x-axes for Sup Fig S7 for awake theta sequence difference rather than awake replay difference would produce the same correlations (as suggested by the data in Fig 7B). Second, these

analyses still require the authors to faithfully identify replay-encoding SPW/Rs, which weakens the argument that these are decoding-independent measures. To address these concerns, I would like the authors to repeat the analyses for Figure S7, but correlate total firing in the task to total firing across all pre-experience SPW/Rs or to total firing across all POST SPW/Rs to rule out the possibility that the observed correlations are simply due to consistent, inherent differences in activity from cell to cell.

Alternatively (or perhaps in addition to the above suggestion), the authors could examine the activity of neurons with place fields in **ONLY** Track1 or Track2. The authors could compare these “track-selective” place cells during run and during POST rest. Assuming that Track1-selective cells have similar place field properties as Track2-selective cells (equivalent place field size, information/spike, firing rate, etc.), the authors should be able to predict differences in the firing of Track1- vs. Track2-selective neurons in POST SPW/Rs that are not trivially explained by on-task firing rates or decoding accuracy.

We agree with the reviewer that the potential confound arising from correlated firing rates across states is critical to address. Indeed when comparing each neuron’s total firing during RUN SWR to total firing across all PRE or POST SWR, we expected the total firing to be correlated. However, we believe it is the relative difference in the firing rates between two tracks (both for behaviour and replay) that matters.

Therefore, based on the reviewer’s suggestion, we decided to demonstrate single-cell-based confirmation of our main finding using a decoding-independent approach. To be specific, we selected SWR events as track 1 or track 2 reactivation events when the proportion of track 1 selective place cells active is 20% greater than the proportion of track 2 selective place cells active (with a track 1 selective cell defined as not having a significant place field on track 2). For example, in a hypothetical situation where there are 10 track 1 selective place cells and 12 track 2 selective place cells for a given session, six active track 1 selective cells and three active track 2 selective cells during a ripple event would correspond to 60% (6/10) and 25% (3/12) of active track 1 and track 2 neurons. This ripple event would therefore be classified as a track 1 event.

While cells only active on a single track are used for replay detection in this analysis, only place cells with place fields on both tracks are used to quantify the difference in firing rate and participation (i.e. number of events a given cell fired at least one spike) between track 1 and track 2 events during PRE, RUN and POST ripple events. This avoids any circular relationship, where detection can directly influence this analysis. By quantifying the predictive relationship between PRE and RUN1, POST1 and RUN1 and POST2 and RUN2, we could demonstrate that the place cell firing (firing rate difference or participation track difference) during awake ripple event could explain firing during POST ripple event but not during PRE ripple event, which is consistent with our finding in Figure S7. We have now included this analysis as Figure S9 and updated the manuscript on page 12 accordingly:

“.....indicating that the more times a cell participates in the replay of a given track, the more likely that cell is to participate during the sleep replay of that track. To confirm that the result is not trivially explained by overall firing rate on the track and during sleep or decoding accuracy, we further applied a decoding-independent analysis. Instead of using Bayesian decoding, we classified SWR events as either track 1 or track 2 reactivation events when the proportion of one of the track’s selective place cells was 20% greater than the proportion from the other track (where a track selective cell only had a place field on one of the tracks). Next, using place cells with place fields on both tracks (a non-overlapping population), we quantified the difference in firing rate and participation between track 1 and track 2 events during PRE, RUN and POST SWR events. We observed that place cell firing (e.g. firing rate difference or participation track difference) during

awake SWR events could explain firing during POST SWR events but not during PRE SWR events (**Fig S9**). These results collectively suggest that the relationship between the number of awake replay and the rate of replay during post-experience sleep is highly conserved for individual neurons, regardless of their magnitude of firing rate on the track or overall level of participation during sleep replay.”

Figure S9. Comparison of the difference in SWR firing rate (top) and SWR event participation (bottom) for PRE vs RUN1 (left), POST1 vs RUN1 (center), and POST2 vs RUN2 (right)

3. In the discussion, I think it is important to expand on the notion of familiarity vs. repetition. It seems to me that familiarity is just repetition with a larger temporal gap between experiences. Do the authors think that repetition is the number of similar experiences during a single waking bout, while familiarity is comparing experiences on opposite ends of sleep/rest periods? How much time do the authors think defines “repetition” vs. “familiar”? Given the opposite network effect of these two conceptually similar terms, it is worth expanding on this notion in a paragraph or so. For example, are there known molecular/synaptic/circuit changes that might ‘tag’ temporal events for a short time, or do the authors think that sleep or sleep replay ‘resets’ the repetition counter? If experiences have simultaneously different levels of familiarity AND repetition, do the authors think one or the other would be dominant? I realize the discussion would be somewhat speculative in nature, so I don’t expect the authors to go into great detail (indeed, the discussion could be shorter than this paragraph I am writing to explain the matter to be discussed!), but I think this is important to place the work into a broader context.

We agree with the reviewer with the need to clarify the conceptual difference between repetition and familiarity. We would agree with the reviewer’s characterization that repetition is the number of similar experiences during a single waking bout. Although we did not test this, we would speculate that it should not matter if the animal had a break *without sleeping*, and was re-exposed to the same track, where additional laps in the re-exposure would count as additional repetition (and lead to higher rates of sleep replay). Given the rate of replay decrease over sleep, the effects of repetition on the original track’s replay rate should go away after sufficient sleep. *Without* a re-exposure to the

track, the rate of replay should remain low in the second sleep session, such that a new novel experience would replay at a relatively higher rate.

Familiarity is based on how much prior experience the animal had in an earlier exposure on the track, but we do not know the amount of time required between exposure needed, or if sleep is required. The key observation is the decrease in awake replay rate which correlates with the amount of remapping between the first and second exposure, suggesting that more stable representations (across exposures) will have lower rates of awake replay (and less sleep replay, if other parameters, like the duration of the behavioural episode remain unchanged).

As for what molecular/synaptic/circuit changes that can lead to this tag, we would speculate that this is happening at least at a synaptic level (although molecular level changes may also be happening). One possibility is that awake replay reactivates cortical networks associated with the replayed memory in conjunction with neuromodulatory pathways (e.g. VTA-dopamine). This results in the increased excitability of the memory trace within cortical networks, which biases the hippocampus towards replaying the same memory during sleep. With more awake replay, this tag is strengthened, while with repeated replay during sleep, this “tag” is weakened, decreasing the excitability of the memory in cortex, and decreasing the memory’s replay rate as the sleep session progresses.

In our experiments, repetition was varied for novel tracks. We would expect similar results for familiar tracks, except if the rate of awake replay was less, than the rate of sleep replay would also be less (though still co-varying with repetition). It is hard to know whether a very familiar repeated experience would replay during sleep more or less than a novel non-repeated experience. Extrapolating from our observations, it should be based on the cumulative amount of awake replay events. So as long as awake replay doesn’t decrease to zero for the familiar track, it should replay more during sleep with sufficient repetition. We have updated the manuscript to reflect the discussion about the effect of novelty and familiarity on replay on page 14-15.

4. In Fig 1 and Sup Fig 1, the authors quantify that total moving time, immobility, and moving speed are equivalent between tracks 1 and 2 in RUN2. However, the important behavioral measure here is likely total experience, which is more directly quantified by number of laps or total distance covered (particularly important since in RUN2 the rats get a fixed amount of time rather than a fixed number of laps). Given the similarity in velocity and time spent mobile, it is very likely that total distance covered and total number of laps will be similar between tracks 1 and 2 in RUN2, but I believe this is important to quantify in order to make this point clear and facilitate proper interpretation of the core finding.

The reviewer makes an excellent point that total distance and number of laps should be quantified, especially for RUN2. We have now extended the signed rank test to the total number of laps ran on each track for each exposure. This analysis could confirm that the total experience on track 1 was significantly greater than that on track 2 during RUN1 (**two-tailed Wilcoxon signed rank test $p = 0.0001$, $n = 19$**), but the total experience on both tracks were equivalent during RUN2 (**two-tailed Wilcoxon signed rank test $p = 0.72$, $n = 19$**). We have included this analysis into Fig 1 and S1 and updated the manuscript on page 2 accordingly:

“During RUN1, the additional laps on track 1 relative to track 2 (Fig 1C and Fig S1A), led to an increase in the total time spent immobile (speed < 4cm/s) (Fig 1D and Fig S1B, two-sided Wilcoxon signed-rank test, $p = 0.00013$, $n = 19$) and running (Fig 1E and Fig S1C, two-sided Wilcoxon signed-rank test, $p = 0.00013$, $n = 19$). However, no difference was observed between the two tracks during the re-exposure session (RUN2, two-sided Wilcoxon signed rank test, $p = 0.78$ for total time spent

immobile and $p = 0.26$ for time spent mobile, $n = 19$). Furthermore, across all protocols, the average running speed of the animal between tracks remained similar (Fig 1F and Fig S1D, two-sided Wilcoxon signed-rank test, $p = 0.94$ for RUN1 and $p = 0.97$ for RUN2, $n = 19$).“

5. The definition of sleep is a little unclear. If the rat moved less than 4 cm/s for a window of 60s, was it assumed to be asleep for the entire window? It seems more likely that rats will be immobile but still awake for a short time before falling asleep, and therefore it may be better to exclude some duration (perhaps the first 30-60 s) of immobility from sleep quantification. Please clarify this in the methods. In addition, given the absence of EMG measurements, the authors should refer to this as putative sleep.

The quantification of sleep was performed independently for each 60s time bin. If the rat moved less than 4cm/s for a window of 60s, this immobility window would be considered a candidate sleep window. Therefore, there will be cases where the rats might be still awake for a small fraction of the immobility windows. However, the subsequent z-scored MUA threshold was applied to exclude immobility windows with large brain state variability. In addition, due to the fragmented sleep patterns in rats, it is difficult to precisely define onset and offset of each sleep epoch in our study, and for these reasons we have avoided removing the first 30-60 seconds from our analysis.

We agree with the reviewer that the ‘sleep’ state in our study should be more conservatively referred to as ‘putative sleep’ due to its relatively relaxed criteria. However, after defining sleep replay as “replay during putative sleep” when first introduced, and emphasizing why we are using the term “putative”, we would prefer to use the term ‘sleep replay’ for brevity afterwards. We have updated the manuscript on page 4-5 and page 25 to reflect this:

Result section

“Replay events during POST1 and POST2 were classified as replay during putative sleep state (referred as sleep replay hereafter) if the replay occurred when the animals’ mean moving speed within a one-minute time bin was lower than 4cm/s and z-scored MUA activity of the most active units (top one-third of the units) was above 0, otherwise they were classified as rest replay (see Methods).”

Method section

“Putative sleep was defined by periods of immobility (windows of 60 s with velocity lower than 4cm/s) accompanied by transient periods of high multi-unit activity (z-score greater than 0). In order to reduce noise levels when setting the multi-unit activity (MUA) threshold, only the most active units (top 1/3 units in terms of total spike counts) were used. To account for intrasubject variability, both the velocity and MUA threshold were visually checked for each data and session and corrected if needed. As a control, we compared our putative sleep detection method with one based on the LFP activity, as it has been commonly used in other studies (Giri et al., 2019; Muessig et al., 2019). For each session, we calculated the LFP theta/delta ratio based on the channels with higher theta and delta power, respectively. Awake periods were classified as high z-scored theta power (>0.5) and high velocity (>4 cm/s). NREM periods were characterized by low z-scored theta power (<0.5), low mobility (<4 cm/s) and high z-scored theta/delta ratio (>0.5), while REM sleep periods were classified as high z-scored theta power (>0.5), low mobility (<4 cm/s) and high z-scored theta/delta ratio (>0.5). For the purposes of this analysis, NREM and REM periods were merged, and then compared to the putative sleep periods obtained with our detection method. The putative sleep detected periods with each method were highly correlated (mean all sessions = 0.51 ± 0.15 , with a mean p -value $< .001 \pm .002$).”

6. Minor Issues:

The figure legend for Fig S4 (C,D) says “Cumulative sleep replay”, but I believe that should be “Cumulative rest replay.”

Thanks for catching the error. We have fixed this typo in the updated manuscript. Please note that the Fig S4 is now Fig S5 in the new manuscript.

In the Methods, please state the total number or percentage of the “few occurrences” in which replay events were significant for both tracks to allow assessment of the accuracy/precision of the decoding.

The mean proportion of replay events significant for both tracks during first exposure and re-exposure were 0.0431 ± 0.0048 and 0.0571 ± 0.0047 , respectively. We do not think the proportion of multi-tracks events is low enough to have any major impacts on our findings. We have updated the manuscript to include this information in the Methods section on page 26- *“In a few occasions, replay events were found to be significant for both tracks (hereafter referred as multi-tracks events). The mean proportion of multi-tracks events during first exposure and re-exposure were 0.0431 ± 0.0048 and 0.0571 ± 0.0047 , respectively. Those events were assigned to one of the tracks by computing the “Bayesian bias” score for each track. Each score was calculated as the sum of the posterior probability matrix for one track, normalized by the total sum across tracks. To assign the replay event to a specific track, the Bayesian bias score was required to be greater than 60%, otherwise the event was discarded. We do not think the proportion of multi-track events is low enough to have any major impacts on our findings.”*

The authors restrict MUA activity analyses to “only the most active units.” For purposes of replication, please describe the criteria used for selection here.

MUA analysis was restricted to top 1/3 of units in terms of total spike counts to reduce the influence of noise or neurons with low firing rates on MUA thresholds. We have updated this information in the Methods section:

“In order to reduce noise levels when setting the multi-unit activity (MUA) threshold, only the most active units (top 1/3 units in terms of total spike counts) were used.”

In the final sentence of the abstract, I believe that “selectively” should be “selective,” or perhaps the “the” should be removed.

Thanks for catching the error. We have fixed this typo in the updated manuscript.

In Sup Fig S3, should the regression b value in panel C be negative?

Thanks for catching the error. We have fixed this typo in the updated manuscript.

Reviewer #2 (Remarks to the author):

1. My main concern is that the protocol covaries # trials and # rewards: because of increased runs on T1, the rats receive more reward on T1. Could all the results be explained by this, meaning the findings are similar to those of Singer & Frank

(<https://pubmed.ncbi.nlm.nih.gov/20064396/>)? I do not see how repetition and reward can be disentangled using this paradigm.

We agree with the reviewer that the number of laps ran and the amount of total physical reward received can be correlated in many situations. In particular, for RUN1 where both tracks were similar in terms of their novelty, the repetition of laps and cumulative amount of physical reward are indeed difficult to disentangle. However, while we can measure reward physically, it is its subjective effect on memory that ultimately we need to measure, which is challenging. For example, increased satiety with additional laps may cause a decreasing reward value. Furthermore, we suspect that if the task was performed without reward, the sheer novelty of the track could act as a reward signal (with longer duration experiences having a larger cumulative reward). Nevertheless, we agree that it is important to indicate in the manuscript that the total experience (i.e. number of laps) may correlate with the cumulative reward, both the physical reward and novelty of the track, and have discussed this alternative view in the revised manuscript on page 14.

“Furthermore, because the animal received a reward for each spatial trajectory, it is difficult to disambiguate between their direct relationship with replay rate. While physical reward can be measured, the subjective value of the spatial trajectory (which may change with novelty and satiety) is challenging to evaluate, and may even change across the behavioral episode.”

In the Singer and Frank study, the rewarded portion of the task led to more awake replay than the unrewarded portion of the task. This may be a similar effect to what we observe with familiarity (e.g. the more novel track had higher replay rates). However, the Singer and Frank study did not look at the cumulative effect of reward, nor did they look at the effect on sleep replay, in contrast to our study. It is worth pointing out that while the cumulative effect of reward (or number of laps) is influencing sleep replay, it does not change the rate of awake replay during the task. Collectively, these results seem to indicate that our findings are distinct but complementary to the observation made by **Singer and Frank (2009)**.

2. The behavioral analyses indicate that rats pause more and move more on T1 than T2 during RUN1, but that running behavior on T1 and T2 is similar during RUN2. Since the rats do not show discrimination between T1 and T2 on RUN2 re-exposure, one interpretation is that the increased prior exposure to T1 has no significant bearing on their behavior, i.e. their memory. I certainly think that extrapolating to “perceived future relevance” is a stretch.

The behavioural analyses were meant to demonstrate that the difference between track 1 and track 2 sleep replay were explained by the total experience on each track during RUN1 but not RUN2. We acknowledge that the behavioural paradigm is not designed to capture behavioural readout of the contextual discrimination, rendering claims about the behavioural significance or relevance speculative. However, we still think that the major difference between the two exposures is the presence of prior experience (or familiarity), which clearly had the impact on awake and sleep replay and theta sequence during re-exposure sessions. To further support our argument, there is evidence from fMRI data showing that human hippocampal reactivation during post-learning rest prioritizes weakly learned information and predicts subsequent memory performance (**Schapiro et al., 2018**). Nevertheless, we are happy to update the manuscript to avoid too many speculative claims about animal’s perceived behavioural significance due to lack of direct behavioural readout. In particular, for the sentence mentioned by the reviewer, we have modified the expression on page 15 as follows:

"Together these data suggest that a memory will be prioritized to replay more during sleep, if there has previously been more awake replay of this memory, cumulatively over the behavioral episode, with repetition of an experience and novelty of the context factors that can generate more awake replay. "

3. What is the nature of the running and pausing behaviors (particularly on T1 during RUN1), where on the track do awake replay events occur and which portions of the track are replayed? These details may help to illuminate whether/how familiarity/reward shape coding of the environment under these conditions.

We have now included the analysis showing where on the track do awake replay events occur and which portions of the track are preferentially replayed. In brief, we observed that replay predominantly occurred at both ends of the track, which are also the parts of the tracks that are preferentially replayed. We also confirmed that local context (the track that the animal is physically exploring) was also the most probably context to replay. As similar findings have been reported previously, we have not included these figures in the revised manuscript, but are happy to do so upon the reviewer's request.

4. How large are “large ensembles” and did the # or proportion of units participating in replay events vary systematically with conditions?

We have included a bar plot in a new supplementary figure (i.e. Fig S3A) showing the number of place cells with place fields on track 1 and/or track 2 for each session during RUN1 and RUN2. The majority of sessions had at least 50 place cells per track.

In addition, we have also included the analysis in Fig S3B looking at the proportion of place cells active during replay. We observed that the proportion of place cell active during POST sleep was correlated with the sleep replay rate for both the first and second exposure to the track. However, the proportion of place cells active during RUN correlated with cumulative awake replay number during first exposure but not re-exposure. This suggests that the proportion of active place cells during replay can correlate with changes in replay rate, but this is not always the case. No difference was observed for events detected as significant during the PRE sleep session. This analysis has been added to the revised manuscript as a supplementary figure (Fig S3).

Figure S3. Number of place cells and place cell participation across experiments (A)
 Summary of number of place cells in each recording session for RUN1 (left) and RUN2 (right). Place cells were either selective for track 1 (light red), selective for track 2 (light blue) or had place fields on both tracks (dark red/blue). **(B)** Proportion of active cells during sleep replay events for PRE (left), RUN 1+2 (center), and POST 1+2 (right). (** $p < 0.01$, *** $p < 0.001$, two-tailed Wilcoxon signed rank test).

5. How much sleep and rest was there during POST1 and POST2? Was the 30min of cumulative sleep selected for analysis distributed similarly across time, independent of session type?

We have now added this analysis into Fig S3 to demonstrate that while the total duration of the cumulative sleep can vary, the 30 mins of cumulative sleep we selected for analysis distributed similarly across time, independent of session type. We have updated the manuscript on page 5 accordingly:

“We first examined how the number of laps run during RUN1 influenced the rate of replay (i.e. the number of replay events per unit time) that occurred during POST1, focusing on the first 30 mins of cumulative putative sleep (see Methods). We selected first 30 mins of cumulative putative sleep as this timeframe represents the minimum duration that animals attain in most sessions. We could

confirm that first 30 mins of cumulative sleep was distributed similarly cross time, independent of session condition.”

6. Given such similar results for POST “rest” replay (Fig S6) and “sleep” replay, does sleep matter? It might, in particular for hippocampal-neocortical interactions – but these are not interrogated here.

We argue that remote rest and sleep replay share similar characteristics, but there are also several differences that distinguish two types of replay. Firstly, unlike sleep replay, rest replay during POST1 is sensitive to contextual novelty but not repetition (Fig S4), hinting that remote rest replay may reflect animal’s ongoing cognition due to recency and novelty effect rather than tracking cumulative experience. Furthermore, while it is true that awake replay is the best predictor of both remote rest and sleep replay during POST, the number of theta sequences fail to predict the rate of rest replay during POST. These differences may suggest that rest replay and sleep replay during POST may engage overlapping yet distinct neural dynamics such as hippocampal-cortical interactions. Therefore, we see a benefit in distinguishing two types of POST replay for better interpretation of our results. Of course, based on the interaction between awake replay during RUN and sleep replay during POST, it is possible that a similar interaction is dynamically happening across sleep bouts for awake rest replay during POST and the sleep replay during POST. However, this is beyond the scope of this experiment and is part of the ongoing work in the lab.

7. The final 3 paragraphs of the Conclusions are speculatively related to this study and could be shortened – particularly the sections on cortical interrelationships with hippocampal replay.

Thank you for the suggestion. We have shortened the final 3 paragraphs on page to avoid too many speculative discussion of the study. It now reads:

“A fundamental question remains - how does awake replay “tag” a memory, subsequently leading to its prioritization during sleep replay. Which memory replays during sleep at a given moment is likely a stochastic process, however, the probability of a salient memory replaying could still be increased by the brain. We have previously proposed that this bias can be the direct consequence of the increased excitability of cortical neurons that represent the behavioral episode (Lewis and Bendor 2019). Recent evidence suggests that the neocortex influences what the hippocampus replays during sleep (Ji and Wilson 2007, Bendor and Wilson 2012, Rothschild et al. 2016). During sleep, epochs of increased cortical activity (up states) precede epochs of increased hippocampal activity (frames) during which replay occurs (Ji and Wilson 2007). Furthermore, the content of cortical activity prior to a sharp-wave ripple can be used to predict what the hippocampus will replay (Rothschild et al. 2016). Similarly, presenting a task-related cue such as a sound, presumably directly driving auditory cortex,

will increase the likelihood that the spatial trajectory related to this cue subsequently replays (Bendor and Wilson 2012). Together, these data indicate that cortical activity at the start of a cortical upstate influences which memories can replay. Likewise, if cortical circuits representing a memory have an increased excitability during sleep (specifically during the up state), they may in turn have a more influential “vote” on what the hippocampus replays, prioritizing this memory for consolidation.

If an increase in excitability within a cortical circuit is required to prioritize a memory for replay during sleep, then how does awake replay play a role in this process? One possibility is that awake replay may occur alongside the reactivation of neuromodulatory pathways (e.g. dopamine), to allow the “offline” potentiation of co-activated cortical circuits. This appears to be an important distinction between awake and sleep replay, given that the coordinated replay between the VTA and hippocampus has been observed in awake but not sleeping animals (Gomperts et al. 2015). However, this potentiation is likely temporary, as given that the rate of replay decreases over the cumulative sleep (Fig S4A,B), the potentiating effects of awake replay on sleep replay rate should diminish with sufficient sleep such that a memory progressively loses its priority for replay. This provides a potential feedback mechanism for the memory triage process where the more a memory replays, the more its replay rate is decreased, effectively de-prioritizing a memory that has sufficiently replayed.”

Reviewer 3

1) The authors used number of replay events and rate of replay to quantify sleep (and awake) replay. One suggestion is that they may consider using z-score-normalized number of replay events relative to its cell-identity shuffled distribution. Due to differences in templates or number of candidate events across animals/sessions, averaging among animals or comparing different conditions may not be accurate. Using normalized number of replay events should improve this aspect.

We would like to thank for the reviewer’s suggestion. While data normalization can be very useful, we would argue that using the raw number of replay events is more justified in this case. Ultimately, we want to understand the relationship between the number of awake replay events and the rate of sleep replay, which depend on raw values. The fact that there is a significant regression across experimental conditions and animals suggests that the phenomenon is quite robust. Secondly, in using the raw values, we can still account for any biases across experiments and conditions. For most analyses, we are not averaging among animals or sessions but rather comparing track 1 replay event and track 2 replay event within each session using signed rank test. Secondly, in cases where we use regression, we have applied a mixed effect model to include animal identity as a random effect to account for animal variability. We have previously measured the “false-positive rate” of our replay detection method, and found this to be approximately 5-6%. While this may vary slightly with number of cells, we do not anticipate this altering our results, given that there is no consistent trend across conditions for the number of place fields on each track.

2) The authors considered theta sequences and awake replay as potential mechanisms for prioritizing sleep replay. Although they are correlated in some aspects, they are all phenomena of place cell patterns. There is no direct evidence for a mechanistic link among them. I suggest not to

present one as a mechanism of another, only to examine which is best explained or correlated with which.

We agree with the author that theta sequences and awake replay are the observable phenomena in the brain where the underlying neural mechanisms are not clearly identified yet. Both are correlated in a sense that the expression of both phenomena are postulated to be important for memory performance and they both happened during online and offline brain state. Nevertheless, we consider theta sequences and awake replay as temporally structured place cell patterns during running and immobility, respectively, which engage different network dynamics (theta oscillation vs sharp-wave ripple).

We are happy to emphasize that we are examining which neural correlate best explains sleep replay. We have updated the manuscript to reflect this. We are including some examples here:

For example in abstract section on page 1

*“Furthermore, we find that the cumulative number of awake replay events that occur during behavior, influenced by both the novelty and duration of an experience, predicts which memories are prioritized for sleep replay, and provides a more parsimonious **neural correlate** for the selective strengthening and triaging of memories”*

For example in result section on Page 7

*“We next explored candidate neural **correlates** during behavior that predicted the difference in sleep replay rates observed during both POST1 and POST2.”*

On page 8

“More recently, it has been shown that impaired theta sequences result in degraded sleep replay, suggesting that theta sequences may be necessary for the initial formation of memory traces (Drieu, Todorova and Zugaro, 2018). Alternatively, the repeated ordered firing of place cells during behavior, postulated to be necessary for sleep replay to later occur, could instead be produced during awake replay. This form of replay is qualitatively similar to sleep replay but typically occurs during behavioral episodes and outside periods of locomotion-driven theta activity when the animal is resting, grooming, or consuming reward”

On Page 11

*“Using a simple regression analysis, we found that the time spent on a track due to task manipulation (Fig 7A, $R^2 = 0.347$, $p = 1.3 \times 10^{-8}$), as well as all three candidate **neural correlates** were predictive of the rate of sleep replay “*

REVIEWERS' COMMENTS

Reviewer #1 (Remarks to the Author):

The authors have thoroughly addressed my initial concerns. I believe the manuscript is considerably stronger following these revisions.

Reviewer #3 (Remarks to the Author):

The authors have addressed my questions. I support the publication of this manuscript.

We would like to thank the editors and the reviewers for the time and effort taken in reviewing our manuscript and providing constructive feedback.

Below is our response to the reviewers' comments follows the format in which the points from the reviewer are in *black*, while our responses are in *blue*.

Reviewers' comments

Reviewer #1 (Remarks to the Author):

The authors have thoroughly addressed my initial concerns. I believe the manuscript is considerably stronger following these revisions.

Thank you very much for your positive feedback!

Reviewer #3 (Remarks to the Author):

The authors have addressed my questions. I support the publication of this manuscript.

Thank you very much for your positive feedback!